# G protein-coupled receptor-based thermosensation determines temperature acclimatization of *Caenorhabditis elegans*

Kohei Ohnishi[1,2,3,8], Takaaki Sokabe [4,5,6,7] ✉, Toru Miura[2,3], Makoto Tominaga [4,5,6], Akane Ohta [1,2,3] ✉ & Atsushi Kuhara [1,2,3,7] ✉

Animals must sense and acclimatize to environmental temperatures for survival, yet their thermosensing mechanisms other than transient receptor potential (TRP) channels remain poorly understood. We identify a trimeric G protein-coupled receptor (GPCR), SRH-40, which confers thermosensitivity in sensory neurons regulating temperature acclimatization in *Caenorhabditis elegans*. Systematic knockdown of 1000 GPCRs by RNAi reveals GPCRs involved in temperature acclimatization, among which *srh-40* is highly expressed in the ADL sensory neuron, a temperature-responsive chemosensory neuron, where TRP channels act as accessorial thermoreceptors. In vivo Ca$^{2+}$ imaging demonstrates that an *srh-40* mutation reduced the temperature sensitivity of ADL, resulting in supranormal temperature acclimatization. Ectopically expressing SRH-40 in a non-warmth-sensing gustatory neuron confers temperature responses. Moreover, temperature-dependent SRH-40 activation is reconstituted in *Drosophila* S2R+ cells. Overall, SRH-40 may be involved in thermosensory signaling underlying temperature acclimatization. We propose a dual thermosensing machinery through a GPCR and TRP channels in a single sensory neuron.

Animals adopt multiple machineries to sense temperatures, which is critical for acclimatization and survival. A variety of responsible molecules exist in the nervous system and other tissues[1], among which a subset of transient receptor potential (TRP) channels is well recognized as primary thermosensors that are evolutionarily conserved from nematodes to humans[2–5]. Furthermore, in *Drosophila* larvae, thermotactic behavior is regulated by a signaling cascade that includes rhodopsins, G protein α-subunit (G$_\alpha$), phospholipase C (PLC), and the TRPA1 channel. In this cascade, rhodopsins, which are light-sensitive trimeric G protein-coupled receptors (GPCRs), are thought to mediate temperature sensation[6–8]. However, it remains unclear whether GPCRs are activated by temperature changes and can act as thermoreceptors.

The nematode *Caenorhabditis elegans* is a useful model for high-throughput analysis to explore sensory mechanisms at molecular, cellular, and individual levels because of its simplicity and powerful molecular genetics. Studies have investigated the temperature responses of *C. elegans* with respect to various phenomena, including thermotactic and cold avoidance behaviors[9–12]. Thermotactic behavior depends on a signaling cascade in the AFD sensory neuron, including receptor-type guanylyl cyclases (rGCs), phosphodiesterase (PDE), and cyclic nucleotide-gated channels

[1]Graduate school of Natural Science, Konan University, Kobe, Hyogo 658-8501, Japan. [2]Faculty of Science and Engineering, Konan University, Kobe, Hyogo 658-8501, Japan. [3]Institute for Integrative Neurobiology, Konan University, Kobe, Hyogo 658-8501, Japan. [4]Division of Cell Signaling, National Institute for Physiological Sciences, Okazaki, Aichi 444-8787, Japan. [5]Thermal Biology Group, Exploratory Research Center on Life and Living Systems, National Institutes of Natural Sciences, Okazaki, Aichi 444-8787, Japan. [6]Department of Physiological Sciences, SOKENDAI, Okazaki, Aichi 444-8787, Japan. [7]AMED-PRIME, Japan Agency for Medical Research and Development, Tokyo 100-0004, Japan. [8]Present address: Physiology and Biophysics, Graduate School of Biomedical and Health Sciences (Medical), Hiroshima University, Hiroshima 734-8553, Japan. ✉e-mail: sokabe@nips.ac.jp; o_akaneiro@me.com; atsushi_kuhara@me.com

(CNGCs), in which rGCs are believed to act as thermoreceptors[13]. AWC sensory neurons also regulate thermotactic behavior through G$_\alpha$, rGCs, and CNGCs; however, the thermoreceptor has not been identified till date[12,14]. For cold avoidance behavior, temperature is sensed by the ASER gustatory neuron, in which a kainate-type glutamate receptor, GLR-3, functions as a cold receptor, thereby activating G protein signaling[10].

Cold tolerance and temperature acclimatization in *C. elegans* are useful to investigate temperature sensation[15–17]. For instance, 25 °C- or 20 °C-cultivated wild-type animals die at 2 °C, whereas 15 °C-cultivated animals survive at 2 °C; this phenomenon has been defined as cold tolerance (Fig. 1a)[15]. Cold tolerance is also associated with temperature acclimatization[15,18]. For instance, when 15 °C-cultivated animals are transferred to and maintained at 25 °C for 3–5 h, they become

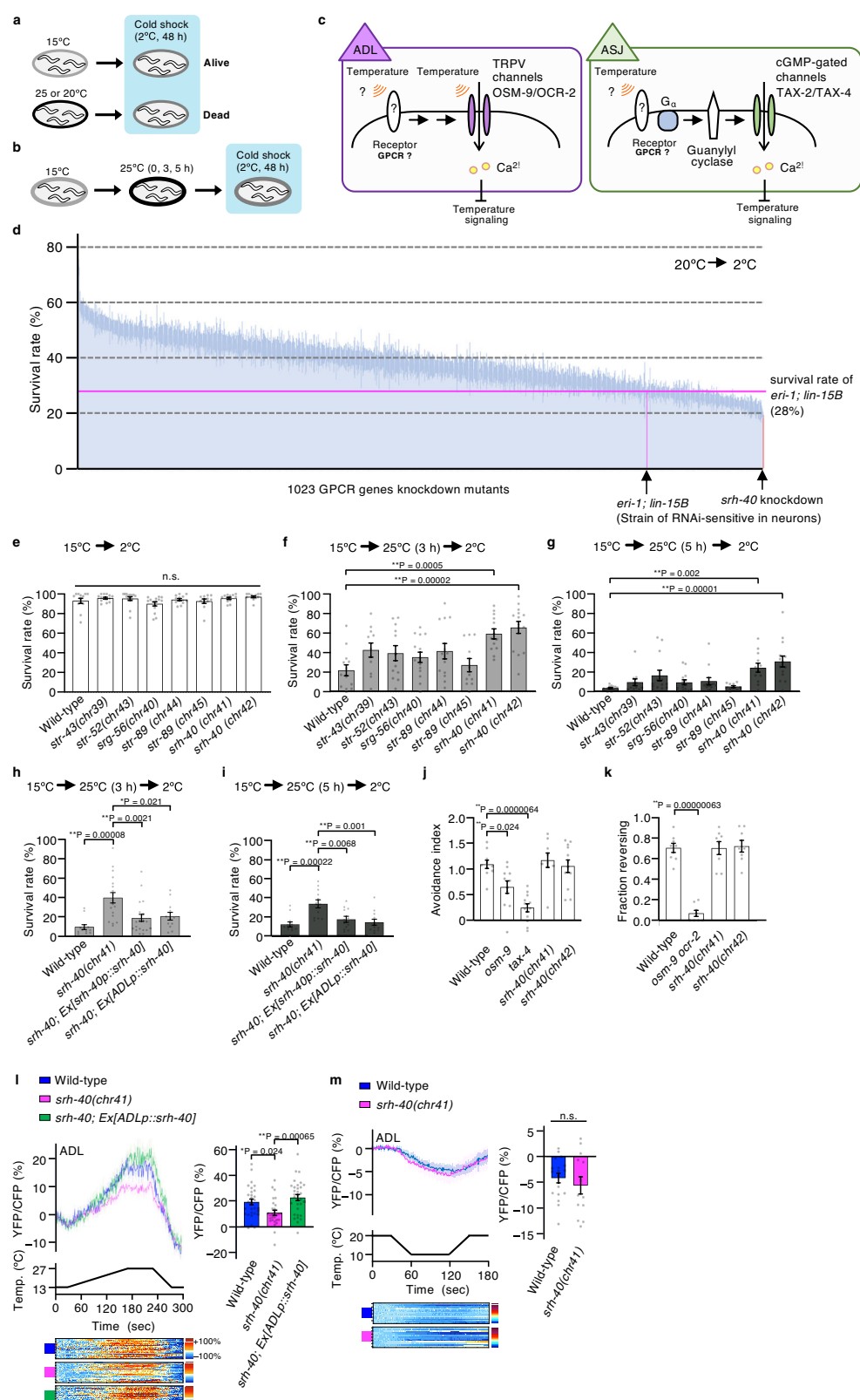

**Fig. 1 | SRH-40 in temperature acclimatization and ADL thermal response.**
**a** Cold tolerance in *C. elegans*. **b** Temperature acclimatization paradigm. **c** Working hypothesis of temperature signaling in ADL and ASJ neurons. **d** Cold tolerance assay for -1000 GPCR genes knockdown animals. *eri-1; lin-15B* (28% survival, a horizontal dotted line) and *srh-40*, as indicated by magenta and red bars, respectively.
**e–g** Temperature acclimatization of GPCR knockout mutants constructed by CRISPR/Cas9 and assayed under [15 °C → 25 °C (0, 3, or 5 h)→2 °C]. *n* = 13, 13, 14, 14, 13, 11, 13, and 12 (**e**), *n* = 13, 11, 12, 15, 13, 11, 13, and 14 (**f**), *n* = 13, 11, 13, 14, 13, 10, 12, and 14 (**g**). **h, i** Temperature acclimatization of the rescued *srh-40* mutants under [15 °C → 25 °C (3 or 5 h)→2 °C]. *n* = 12, 18, 21, and 12 (**h**), or *n* = 12 (**i**). **j, k** Avoidance behavior against 1-octanol (*n* = 10, 11, 12, 9, and 12) (**j**) and pheromone (*n* = 8) (**k**). **l, m** Ca$^{2+}$ imaging of ADL. Traces indicate the averaged YFP/CFP ratio of YC3.60 in response to [13 °C → 27 °C → 13 °C] (**l**; *n* = 30, 29, and 32) and [20 °C → 10 °C → 20 °C] (**m**; *n* = 18, 13). Bar graphs indicate the averaged YFP/CFP ratio between 211–220 s (**l**) and 101–110 s (**m**), where the wild-type showed the maximum changes. Each row in color maps represents relative Ca$^{2+}$ concentration changes from one worm; excluded values > 100% are shown in white. The colors of the bar graphs and color maps correspond to the colors of response traces. *n* indicates independent experiments (shown from left in the bar graph). Bar graphs represent mean ± SEM. *p*-values were calculated using unpaired *t*-test (**m**) or one-way ANOVA with Tukey–Kramer's test (**h, i, l**) or Dunnett's test (**e–g, j, k**). n.s. *P* ≥ 0.05; *$P$ < 0.05; **$P$ < 0.01. All statistical tests were two-sided. Source data are provided as a Source Data file.

intolerant at 2 °C (Fig. 1b). For cold tolerance and temperature acclimatization, ASG, ADL, and ASJ sensory neurons act as temperature-responding neurons (Fig. 1c)[15,19–21]. The sensory neuron ASG regulates cold tolerance through a downstream interneuron, in which DEG-1, a degenerin/epithelial sodium channel (DEG/ENaC), acts as a thermoreceptor[20]. In the chemosensory neuron ADL, three TRP channels, OSM-9/OCR-2/OCR-1, are required for temperature signaling[19,22], and OSM-9/OCR-2 may act as an accessorial warmth receptor downstream of an unknown thermoreceptor (Fig. 1c)[5]. ASJ mediates light and pheromone sensation, in which the sole photoreceptor LITE-1 activates a downstream signaling cascade, including G$_{\alpha}$, GCs, and CNGCs[23]. Temperature information is also transmitted via G protein signaling as in phototransduction; however, LITE-1 is not involved in temperature sensation and cold tolerance[15,24]. Therefore, in ADL and ASJ, we hypothesize that other receptor(s) such as GPCRs sense temperature upstream of the G protein and ion channels (Fig. 1c).

In this study, we isolate a GPCR in *C. elegans* that is involved in temperature acclimatization. Cold tolerance assay demonstrates that SRH-40 regulates temperature acclimatization in the ADL thermosensory neuron, possibly through G$_{\alpha}$ and TRPV channels. Ca$^{2+}$ imaging of SRH-40 in the gustatory neuron ASER and *Drosophila* S2R+ cell suggest that SRH-40 expression confers temperature responses. We propose that both SRH-40 and TRPV channels could independently or dependently play a role in the thermosensing machinery in a single sensory neuron.

## Results

### SRH-40 is involved in temperature acclimatization
We conducted RNAi screening to isolate the GPCR-based thermoreceptors involved in cold tolerance. Because the effects of RNAi are extremely low in the nervous system of wild-type animals, we used the *eri-1; lin-15B* strain, an RNAi-sensitive strain, to knockdown genes in neurons[25]. *C. elegans* possesses -1700 genes encoding GPCRs, of which -400 are considered pseudogenes[26–33]. We knocked down -1000 GPCR genes in the whole body and evaluated the cold tolerance in each impaired animal (Fig. 1d). Because cold tolerance abnormality was strongly observed in ASJ-defective mutants under the [20 °C → 2 °C] condition[15], we applied the same protocol for this RNAi screening. The survival rate of 20 °C-grown *eri-1; lin-15B* animals, the RNAi-sensitive strain, after cold exposure was 28%, which was used as a control for screening (Fig. 1d, Supplementary Data 1). We observed that the majority of the GPCR knockdown led to enhanced cold tolerance, and the minor population showed reduced cold tolerance (Fig. 1d, Supplementary Data 1). Since we have found many genes whose mutations lead to abnormal increases in cold tolerance[15], we focused on genes involved in decreased cold tolerance and further investigated selected genes based on the order of the strength of the abnormality. We evaluated the expression pattern of 53 of those genes using a GFP reporter and DiI staining and found that 16 genes were expressed in temperature-sensing neurons, including ASJ and ADL, in the head (Supplementary Fig. 1, Supplementary Table 1). We then attempted to

introduce mutations in these GPCR genes using CRISPR/Cas9 and evaluated temperature acclimatization of the knockout animals lacking one of 12 GPCR genes under the [15 °C → 25 °C (3 or 5 h)→2 °C] condition (Fig. 1e–g, Supplementary Fig. 2a–c). Among those genes, two independent *srh-40* knockout mutants, *srh-40(chr41)* and *srh-40(chr42)*, consistently exhibited abnormal temperature acclimatization (Fig. 1e–g). *srh-40* expression was detected in the ADL thermosensory neuron (Supplementary Fig. 3a, b), with the expression level being higher at 25 °C than at 15 °C (Supplementary Fig. 3c). Under the [15 °C → 2 °C] condition, both 15 °C-grown wild-type and *srh-40* mutant animals survived comparably after cold stimuli (Fig. 1e, Supplementary Fig. 3d). In contrast, when 15 °C-grown wild-type animals were maintained at 25 °C for 3 or 5 h, -80% or -95% of wild-type animals died after cold stimuli, respectively [15 °C → 25 °C (3 or 5 h)→2 °C] (Fig. 1f, g). However, the survival rates of *srh-40* mutants were significantly higher under the [15 °C → 25 °C (3 or 5 h)→2 °C] condition (Fig. 1f, g), suggesting that SRH-40 is essential for temperature acclimatization. The *srh-40* knockout mutants did not exhibit abnormal cold tolerance under the [20 °C → 2 °C] condition (Supplementary Fig. 3e, f), which was inconsistent with the phenotype of *srh-40* knockdown animals (Fig. 1d). This phenotypic discrepancy between knockout and knockdown might be due to off-target effects of RNAi and/or, e.g., random effectiveness of RNAi among tissues, because of a drastic reduction of *srh-40* expression in the *srh-40* knockdown animals (Supplementary Fig. 3g). Considering that the expression level of *srh-40* was higher at 25 °C than at 15 °C (Supplementary Fig. 3c), SRH-40 should be more important for temperature acclimatization than for cold tolerance.

### SRH-40 is required for the thermal response of ADL neuron
To examine whether the high survival rate of *srh-40* mutants was due to SRH-40 dysfunction in ADL, we expressed *srh-40* cDNA in the ADL neuron of the mutants. The abnormal increase in cold tolerance in *srh-40* mutants was rescued by the expression of *srh-40* cDNA driven by an *srh-40* promoter or an ADL-specific promoter (Fig. 1h, i and Supplementary Fig. 3h, *srh-40; Ex[srh-40p::srh-40] or srh-40; Ex[ADLp::srh-40]*). These findings suggest that SRH-40 in the ADL is required for normal temperature acclimatization. Studies have reported that ADL detects multiple sensory stimuli, such as aversive odorant 1-octanol, aversive ascaroside pheromones, and temperature[19,26,34,35]. However, we found that the *srh-40* mutants exhibited normal avoidance against 1-octanol and aversive pheromones (Fig. 1j, k), suggesting that the SRH-40 expressed in ADL is not involved in the avoidance of 1-octanol and aversive pheromones, and that the disabled temperature acclimatization in *srh-40* mutants is not due to a general defect in the neural functions.

To explore whether SRH-40 mediates temperature sensation in ADL, we performed Ca$^{2+}$ imaging in the ADL using a genetically encoded Ca$^{2+}$ indicator, YC3.60. The Ca$^{2+}$ concentration in the ADL increased upon warming (13 °C → 27 °C) (Fig. 1l), consistent with previous reports[19,22]. The increase in Ca$^{2+}$ concentration upon warming was lower in the ADL of *srh-40* mutants than that in the wild-type animals and recovered to the levels of wild-type animals by the specific

introduction of *srh-40* cDNA in the ADL (Fig. 1l). In contrast, the Ca²⁺ concentration in the ADL slightly decreased upon cooling (20 °C → 10 °C) in both wild-type animals and *srh-40* mutants (Fig. 1m). These findings suggest that SRH-40 is essential for warm rather than cool activation of ADL. The morphological structure of the ADL, especially the axon, dendrite, and cell body, of *srh-40* mutants, was indistinguishable from that of wild-type animals (Supplementary Fig. 3i), implying that SRH-40 is not involved in ADL development.

Despite the reduction in the ADL thermal responsiveness of *srh-40* mutants, as measured using cameleon YC3.60, the response was not completely abolished (Fig. 1l). These data suggest that SRH-40 is not the exclusive thermosensor, and other thermosensor(s) also acts in ADL temperature signaling.

### G$_{\alpha q}$ EGL-30 and TRPV act in ADL temperature signaling

To determine the downstream effectors of SRH-40 in temperature acclimatization, we attempted to identify the G$_\alpha$ and TRP channels involved in ADL temperature signaling. As reported previously, the TRPV channels OSM-9/OCR-2 are involved in temperature sensation in ADL[5], and several TRP channels are regulated by G$_{\alpha q}$ signaling[36]. ADL expresses multiple G$_\alpha$, such as EGL-30 and GOA-1. Because G$_{\alpha q}$ EGL-30 functions upstream of TRPV channels in the ASH sensory neuron that regulates avoidance behavior[37], we evaluated whether G$_{\alpha q}$ EGL-30 and TRPV channels are involved in temperature acclimatization and temperature response of ADL (Fig. 2a–f).

EGL-30 is the sole G$_{\alpha q}$ in ADL, and the *egl-30(ad806)* mutant harbors a reduction-of-function mutation, which led to an abnormal increase in survival rate under the [15 °C → 25 °C (3 or 5 h)→2 °C] condition (Fig. 2a–c left panels). Knockdown of ADL-specific EGL-30 in wild-type animals resulted in a similar cold tolerance phenotype as that of *egl-30(ad806)* (Fig. 2a–c right panels). Ca²⁺ imaging of ADL revealed that animals with impaired EGL-30 in ADL exhibited decreasing temperature responses compared with those of the wild-type neuron (Fig. 2d). However, the specific expression of a constitutive active *egl-30* (Q205L) in the ADL of wild-type animals increased the temperature response compared with that in wild-type animals (Fig. 2e). These data suggest that EGL-30 positively regulates temperature signaling in ADL. ADL also expresses G$_{\alpha i/o}$ encoded by *goa-1*, but *goa-1* mutant demonstrated normal temperature responses (Supplementary Fig. 3j). To explore the functional relationship between EGL-30 and SRH-40 in ADL, we measured the ADL temperature response in wild-type animals overexpressing SRH-40 in ADL with or without an ADL-specific EGL-30 knockdown (Fig. 2f). SRH-40 overexpression in ADL led to an augmented temperature response in ADL compared with that in wild-type animals, and this response was suppressed by ADL-specific EGL-30 knockdown, which was comparable to the level observed with ADL-specific EGL-30 knockdown alone (Fig. 2f). These data suggest that EGL-30 functions downstream of SRH-40 in ADL temperature signaling.

Previous studies have demonstrated that TRPV mutants exhibit defects in temperature acclimatization and decreased ADL temperature responses, suggesting that TRPV channels are crucial components in ADL temperature sensing[5,19,22]. Furthermore, *osm-9* and *ocr-2* single mutants and *osm-9 ocr-2* double mutant exhibited abnormal temperature acclimatization under the [15 °C → 25 °C (3 or 5 h)→2 °C] condition, and *osm-9* and *ocr-2* single mutants exhibited decreased ADL temperature responses[5]. However, the *osm-9 ocr-2* double mutant and the *osm-9 ocr-2; ocr-1* triple mutant harboring an additional TRPV mutation exhibited normal ADL temperature responses, possibly because of a compensatory mechanism[5]. To explore the functional interaction between SRH-40 and TRPV channels in ADL, we constructed mutants harboring *srh-40* and multiple TRPV channels and evaluated their temperature acclimatization and ADL temperature responses (Fig. 2g–i). We observed that *srh-40; osm-9 ocr-2* triple mutants demonstrated an abnormal increase of survival rate under the

[15 °C → 25 °C → 2 °C] condition; this phenotype was comparable to that of the *osm-9 ocr-2* double mutant and greater than that of the *srh-40* mutant (Fig. 2g). However, the ADL temperature responses in the *srh-40; osm-9 ocr-2* triple mutant and *srh-40; osm-9 ocr-2; ocr-1* quadruple mutant were indistinguishable from those of wild-type animals (Fig. 2h), which was probably due to an unidentified compensatory mechanism[5]. Because the ADL temperature responses of the TRPV double and triple mutants were not reduced, and the *ocr-2* single mutation was more effective than the *osm-9* mutation in reducing ADL temperature responses[5], we investigated an *srh-40; ocr-2* double mutant. This mutant exhibited decreased ADL temperature responses (Fig. 2i), a phenotype that was comparable to that of the *ocr-2* and *srh-40* single mutants. We also evaluated cameleon YC4.12[38] with a lower Ca²⁺ affinity than YC3.60 and confirmed that the thermal response of ADL was minimal (Fig. 2h), suggesting that the temperature-dependent intensity change of YC3.60 in ADL was due to an increase in Ca²⁺ concentration, but not due to an artifact of the cameleon (Fig. 2h).

Overall, these data suggest that SRH-40, EGL-30, and TRPV channels are crucial components in ADL for temperature acclimatization, and SRH-40 functions upstream of EGL-30. TRPV channels could be downstream of SRH-40 or act in parallel with the SRH-40 pathway.

### SRH-40 confers warmth response in taste and cultured cells

To determine whether SRH-40 itself is sufficient to respond to warming, we ectopically expressed SRH-40 in the right ASE gustatory neuron (ASER). Because ASER does not react to warming stimuli, it has been used to determine the temperature sensitivity of various receptors in *C. elegans*, such as rGCs, DEG/ENaC mechanoreceptors, and TRPV channels[5,13,20]. We expressed GCaMP8 together with SRH-40 in the ASER to detect changes in Ca²⁺ concentration to warming stimuli (13 °C–27 °C). Although naïve ASER did not respond to warming stimuli, the ASER with ectopic expression of SRH-40 exhibited a warming stimuli-dependent increase in Ca²⁺ concentration (Fig. 3a). As ASER responds to cooling stimuli via the cold receptor GLR-3[10], we also evaluated an ASER harboring a *glr-3* mutation. We observed that *glr-3* ASER similarly did not respond to warming stimuli, whereas SRH-40-expressing *glr-3* ASER gained sensitivity to warming stimuli (Fig. 3b). These findings suggest that SRH-40 is sufficient to confer temperature sensitivity to non-temperature-sensitive neurons. We observed that the SRH-40-induced temperature responsiveness of ASER was suppressed by the mutation of G$_\alpha$ encoded by *goa-1*, *gpa-3*, and *egl-30* that are expressed in ASER and ADL thermosensory neurons (Fig. 3c, d). *gcy-5* and *tax-4* encode an rGC and CNGC, respectively, and are required for ASER gustatory signaling[39–41]. Ectopic expression of SRH-40 in ASER in *gcy-5* and *tax-4* mutant backgrounds failed to demonstrate Ca²⁺ response to warming stimuli (Fig. 3e, f), suggesting that SRH-40 uses GCY-5 and TAX-4 to react to temperature stimulation when mis-expressed in ASER; furthermore, TAX-4 may be a primary channel responsible for the increase in Ca²⁺ concentration.

To further examine the temperature responsiveness of SRH-40, we used *Drosophila* Schneider 2 R+ (S2R+) cells as a heterologous expression system for Ca²⁺ imaging. This cell line fulfilled our requirement because (1) S2R+ cells can be maintained at 25 °C, a temperature almost similar to that for *C. elegans*; (2) *srh-40* is well expressed in S2R+ cells (Supplementary Fig. 4a) but hardly expressed in human embryonic kidney 293 T (HEK293T) cells; and (3) co-expression of SRH-40 and G$_\alpha$ causes death of *Xenopus* oocytes. We first evaluated the temperature response of OSM-9/OCR-2 in S2R+ cells, because these TRP channels are required for heat responses in ADL and act as accessorial warmth sensors[5,19,22]. We observed heat stimulation-induced increases in intracellular Ca²⁺ (Ca²⁺$_i$) concentrations in control cells, probably due to the intrinsic temperature sensitivity of a Ca²⁺ indicator Fura-2[42] and S2R+ cells. In contrast, cells expressing OSM-9/OCR-2 exhibited further increases in Ca²⁺$_i$ concentration compared with the control during the first and second heat

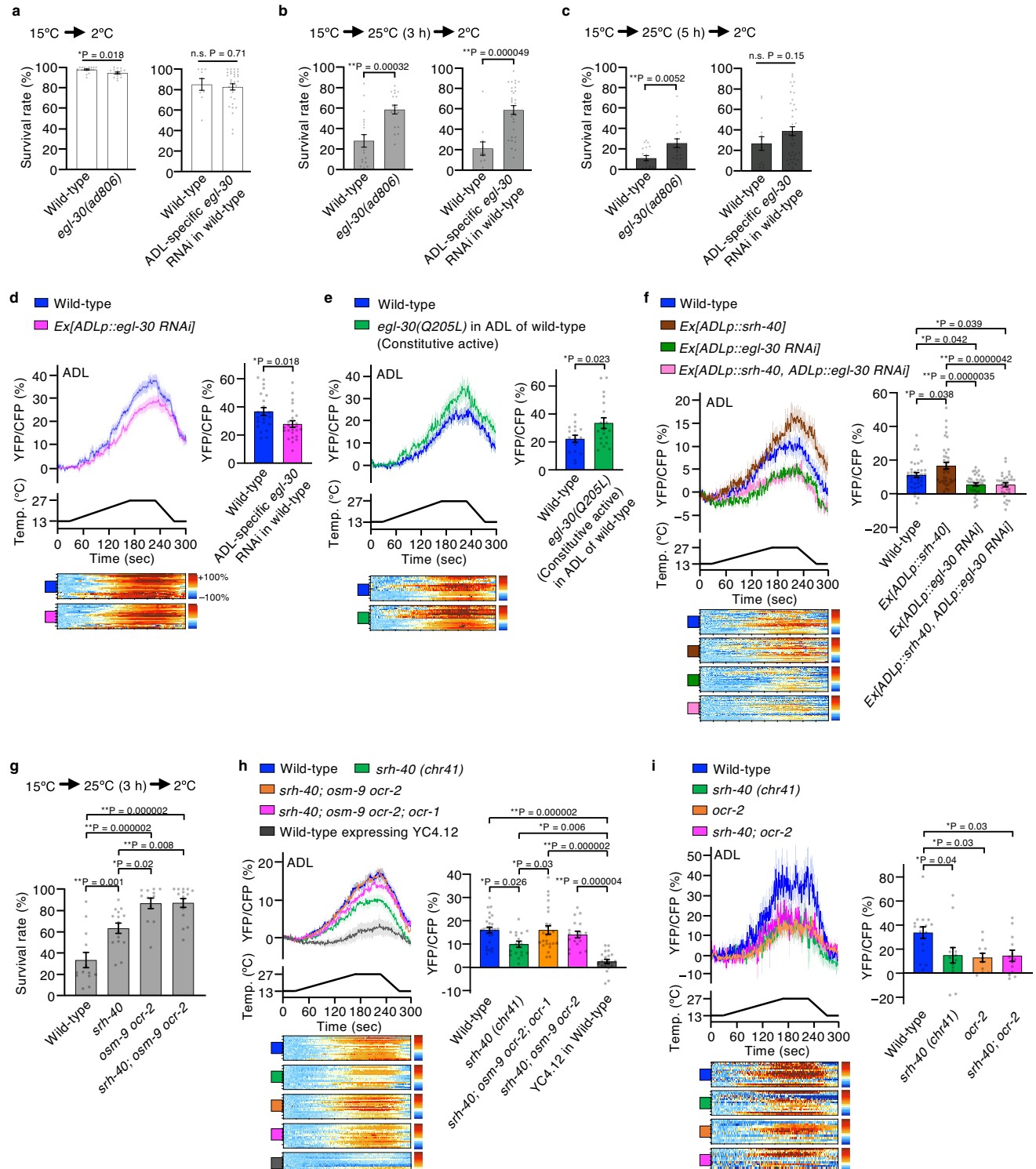

**Fig. 2 | Temperature acclimatization and Ca²⁺ imaging in Gα and TRPV channel mutants in ADL.** **a**–**c** Temperature acclimatization of *egl-30(ad806)* mutants under [15 °C → 25 °C (0, 3, or 5 h)→2 °C] (left panels; $n = 18, 15$ (**a**), $n = 18, 17$ (**b**), $n = 18, 17$ (**c**)). Temperature acclimatization in ADL-specific *egl-30* knockdown animals under [15 °C → 25 °C (0, 3, or 5 h)→2 °C] (right panels; $n = 9$, and 32 (**a**), $n = 11$, and 31 (**b**), $n = 14$, and 42 (**c**)). **d**, **e** Ca²⁺ imaging of ADL neurons, ADL-specific EGL-30 knockdown animals (**d**; $n = 22, 24$), and ADL-specific EGL-30(Q205L) expressing animals (**e**; $n = 16, 19$). **f** Ca²⁺ imaging of ADL in wild-type overexpressing SRH-40 in ADL with or without ADL-specific EGL-30 RNAi. $n = 37, 38, 37$, and 32. **g** Survival rate after temperature acclimatization [15 °C → 25 °C (3 h)→2 °C] in mutants as indicated ($n = 13, 15, 12$, and 15). **h** Ca²⁺ imaging of ADL in mutants as indicated ($n = 27, 17, 25, 18$, and 22). **i** Ca²⁺ imaging of ADL in mutants as indicated

($n = 16, 13, 11$, and 13). In the Ca²⁺ imaging (**d**–**f**, **h**, **i**), traces indicate the averaged YFP/CFP ratio of YC3.60 in response to warming and cooling. The averaged YFP/CFP ratio of YC4.12 is shown in (**h**). Bar graphs indicate the averaged YFP/CFP ratio between 211–220 s, the maximum point in each strain. Each row in color maps represents relative Ca²⁺ concentration changes from one worm; excluded values > 100% are shown in white. The colors of bar graphs and color maps corresponds to the colors of response traces. $n$ indicates independent experiments (shown from left in the bar graph). Bar graphs represent mean ± SEM. *p*-values were calculated using unpaired *t*-test (**a**–**e**) or one-way ANOVA with Tukey–Kramer's test (**f**–**i**). n.s. $P \geq 0.05$; *$P < 0.05$, **$P < 0.01$. All statistical tests were two-sided. Source data are provided as a Source Data file.

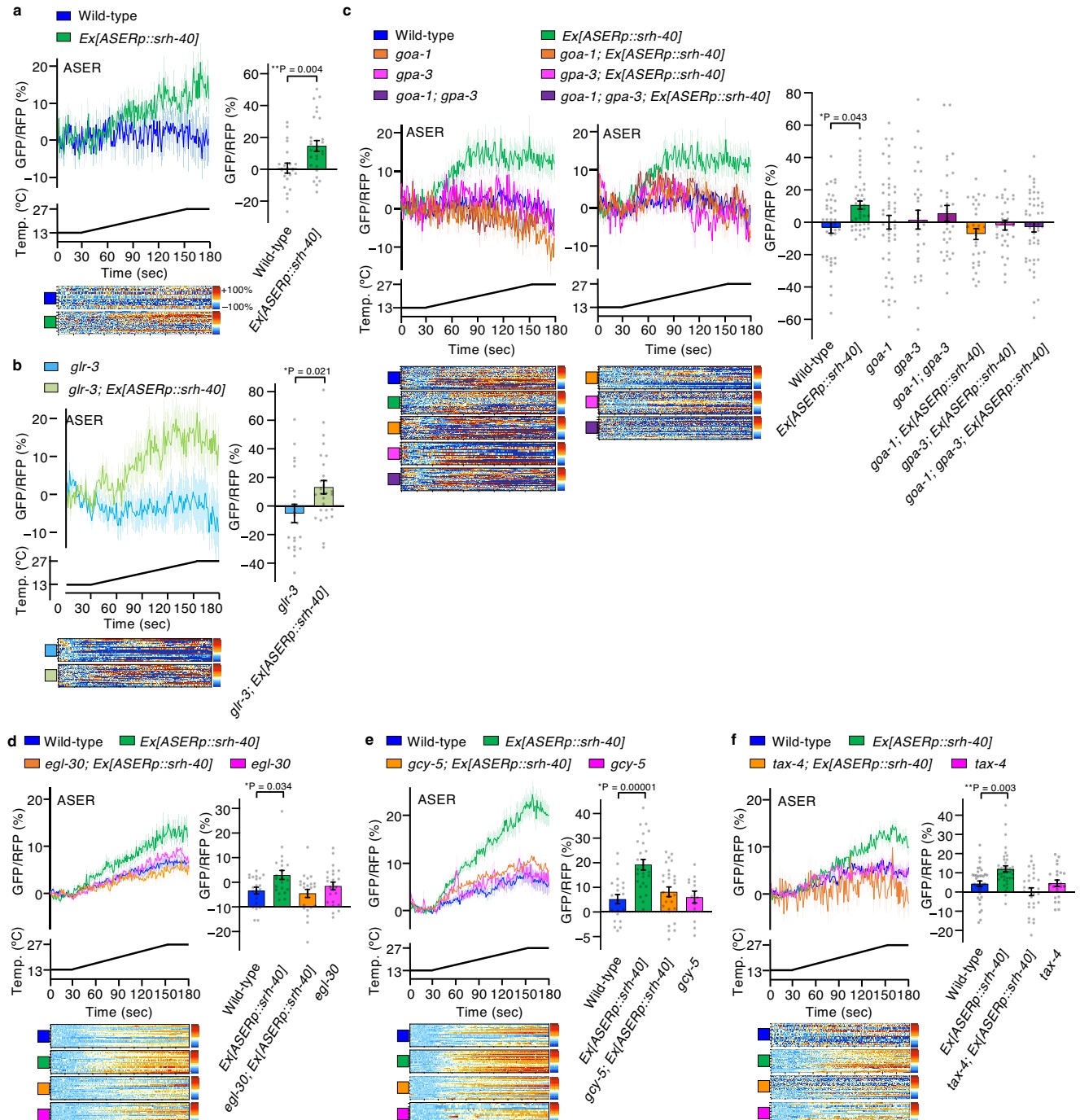

**Fig. 3 | Ca²⁺ imaging of ASER gustatory neuron expressing SRH-40 in various *C. elegans* mutants.** Traces indicate Ca²⁺ concentration changes in the ASER gustatory neuron expressing SRH-40 upon warming stimulation in wild-type (**a**; $n = 16$, 19), *glr-3* mutant lacking a cold receptor (**b**; $n = 21$, 28), *goa-1* and *gpa-3* mutants lacking $G_\alpha$ (**c**; $n = 40$, 41, 25, 30, 43, 49, 22, and 32), *egl-30* mutant lacking $G_\alpha$ (**d**; $n = 20$, 20, 23, and 22), *gcy-5* mutant lacking guanylyl cyclase (**e**; $n = 21$, 26, 27, and 13), and *tax-4* mutant lacking cGMP-gated channel (**f**; $n = 35$, 38, 28, and 14). The trace panels in (**c**) are divided into two for visibility; the traces of wild-type and wild-type expressing SRH-40 are shared between the left and right panels. Traces in the upper panels indicate the averaged GCaMP8/tagRFP ratio in response to warming stimulation. Each row in the color maps indicate relative changes in Ca²⁺ concentrations in individual neurons. Bar graphs indicate the averaged GCaMP8/tagRFP ratio between 161–180 s, where the temperature reached the maximum value. *n* indicates independent experiments (shown from left in the bar graph). Bar graphs represent mean ± SEM. *p*-values were calculated using unpaired *t*-test (**a**, **b**), one-way ANOVA with Dunnett's test (**e**), or Kruskal–Wallis's test with Steel's test (**c**, **d**, **f**). n.s. $P \geq 0.05$; *$P < 0.05$, **$P < 0.01$. All statistical tests were two-sided. Source data are provided as a Source Data file.

stimulations (Fig. 4a–c), suggesting that these TRPV channels are responsive to temperature increases as reported previously[5], and that the Ca²⁺ response of the thermoreceptor is measurable in S2R+ cells.

Therefore, we constructed an SRH-40-stable cell line in which *srh-40* was integrated into the genome to evaluate the temperature response of SRH-40. *srh-40* expression was induced by copper (CuSO₄), and other components were transiently transfected. Cells that were not treated with CuSO₄ and transfected with a vector plasmid were used as the negative control. To equalize the transfection condition among samples, we transfected the same amount of a DNA mixture (1.2 μg in total) to cells in all samples. The heating-induced increases in Ca²⁺ᵢ concentration in cells expressing OSM-9/OCR-2 with

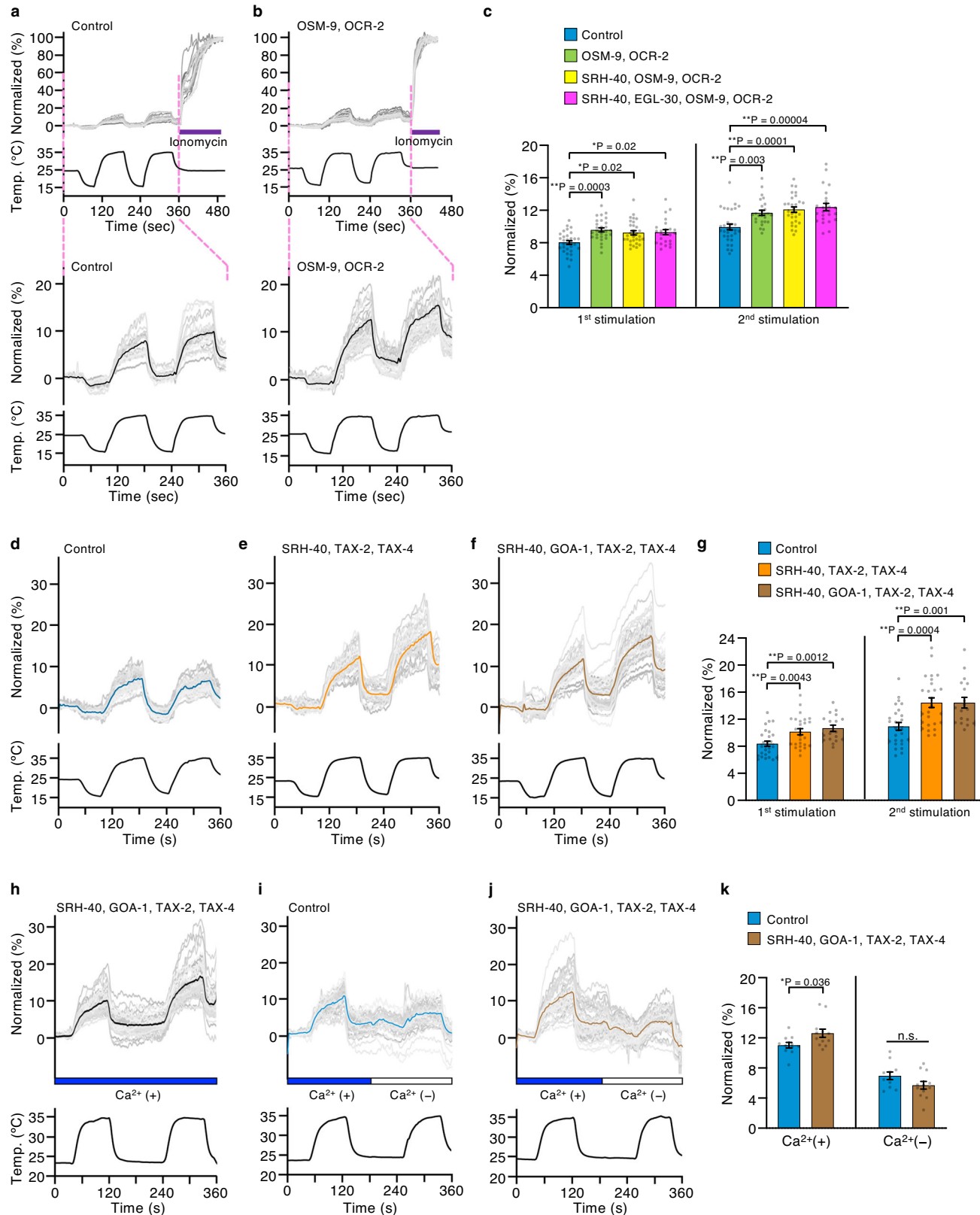

SRH-40 and SRH-40/$G_{\alpha q}$ EGL-30 were significantly higher than those in the control but comparable to those in cells expressing only OSM-9/OCR-2 (Fig. 4c). This finding suggested that these components were not sufficient to monitor the temperature response of SRH-40.

We then introduced $G_\alpha$ GOA-1 and CNGC TAX-2/TAX-4 into the SRH-40 stable S2R+ cell line, because we could observe a temperature-

dependent $Ca^{2+}$ increase with ectopic SRH-40 expression in ASER, and SRH-40 utilized $G_\alpha$ GOA-1 and CNGC TAX-2/TAX-4 to react to the warming stimulation in ASER (Fig. 3). We expressed SRH-40, TAX-2, and TAX-4 with or without GOA-1 in S2R+ cells. Consequently, the $Ca^{2+}_i$ concentration in cells expressing SRH-40/TAX-2/TAX-4 and SRH-40/GOA-1/TAX-2/TAX-4 significantly increased during the first and second

**Fig. 4 | Ca²⁺ imaging of *Drosophila* S2R+ cells expressing SRH-40. a, b** Representative traces of individual cells (Gray) and average of Ca²⁺ᵢ concentrations (Black) upon temperature fluctuations in control cells (**a**) and S2R+ cells expressing OSM-9/OCR-2 (**b**). The experiment overview (upper), and expanded view of the thermal stimulation period from 0–360 s (lower, highlighted in pink). **c** Comparison of the maximum increase in Ca²⁺ᵢ concentration in response to first (left) and second (right) heat stimulation in the control and cells expressing various genes as indicated. $n = 1089, 844, 872$, and 697 (1089, 805, 839, and 697 in second stimuli) cells examined over 31, 28, 30, and 24 (31, 26, 29, 24 in second stimuli) independent experiments. **d–f** Representative traces of individual cells and average of Ca²⁺ᵢ concentrations upon temperature fluctuations in control (**d**), cells expressing SRH-40/TAX-2/TAX-4 (**e**) or SRH-40/GOA-1/TAX-2/TAX-4 (**f**). **g** Comparison of the maximum increase in Ca²⁺ᵢ concentrations in response to the first and second heat stimulation in the cells shown in **d–f**. $n = 961$ (920 in second stimuli), 810, and

628 cells examined over 28 (27 in second stimuli), 26, and 18 independent experiments. **h–j** Representative traces of individual cells and the average of Ca²⁺ᵢ concentrations upon temperature fluctuations in a bath containing 2 mM Ca²⁺ (blue bars between the upper and lower panels) or no extracellular Ca²⁺ (white bars between the upper and lower panels) in cells expressing SRH-40/GOA-1/TAX-2/TAX-4 (**h, j**) and control cells (**i**). **k** Comparison of the maximum increase in Ca²⁺ᵢ concentration in response to heat stimulation in the presence (left) or absence (right) of extracellular Ca²⁺. $n = 437, 478$ cells examined over 11, 12 independent experiments. $p$-values were calculated using Mann–Whitney $U$ test (**k**), one-way ANOVA with Tukey–Kramer's test (**g**, and second stimuli in **c**), and Kruskal–Wallis's test with Steel–Dwass's test (first stimuli in **c**). $n$ indicates from left in the bar graph. Bar graphs represent mean ± SEM. n.s. $P \geq 0.05$; *$P < 0.05$, **$P < 0.01$. All statistical tests were two-sided. Source data are provided as a Source Data file.

heat stimulations (Fig. 4d–g). However, such an increase in Ca²⁺ᵢ concentration was absent in cells expressing SRH-40 alone, TAX-2/TAX-4 alone, SRH-40/GOA-1, or GOA-1/TAX-2/TAX-4 (Supplementary Fig. 4b, c), suggesting that the expressions of SRH-40 and TAX-2/TAX-4 were essential for the temperature response.

To determine whether the increase in Ca²⁺ᵢ concentration was mediated by a Ca²⁺ influx via TAX-2/TAX-4, we applied temperature stimuli in the presence or absence of extracellular Ca²⁺ in the bath solution. We confirmed that the responses to the first and second heat stimulations were clearly observed in the presence of extracellular Ca²⁺ (Fig. 4h). Such an increase in Ca²⁺ᵢ concentration was significantly higher in cells expressing SRH-40/GOA-1/TAX-2/TAX-4 than that in the control (Fig. 4k, Ca²⁺(+)). However, when extracellular Ca²⁺ was depleted during the second heat stimulation in the same set of cells, the increase in Ca²⁺ᵢ concentration diminished in cells expressing SRH-40/GOA-1/TAX-2/TAX-4, and the levels were comparable with those in the control (Fig. 4i–k). These data suggested that the heat-dependent increase in Ca²⁺ᵢ concentration in these cells was evoked by SRH-40 activation, followed by Ca²⁺ influx via TAX-2/TAX-4. These results in the heterologous expression system were consistent with the evidence that ectopically expressing SRH-40 in warmth-insensitive ASER conferred thermal responsiveness through Gα and CNGCs and that the ADL temperature response was driven by SRH-40.

## Discussion

Our study describes the function of GPCR SRH-40 required for thermal response of thermosensory neuron in the temperature acclimatization of *C. elegans*. Based on genetic and Ca²⁺ imaging analyses, our results elucidated that SRH-40 and downstream Gα signaling are required for ADL thermosensation and subsequent temperature acclimatization. As the ectopic expression of SRH-40 in the non-warmth-sensing gustatory neuron ASER and *Drosophila* S2R+ cells resulted in the acquisition of warmth sensitivity, SRH-40 could react to temperature changes; however, other possibilities cannot be excluded, such as stabilizing, localizing, and/or facilitating the expression of other thermosensitive molecules that are associated with temperature acclimatization.

In RNAi screening, the knockdown of numerous GPCR genes in animals were consistently associated with an elevation in cold tolerance. This observation aligns with the previous studies, where the majority of mutants isolated previously exhibited abnormal increases, rather than decrease, in cold tolerance[15,19–21,43]. For example, the *unc-104* mutant, impairing a kinesin in most neurons, displayed an abnormal increase in cold tolerance[15]. This implies that a defect in the entire nervous system leads to an enhanced cold tolerance. Moreover, abnormalities in various tissues such as muscle, intestine, and sperm were associated with abnormal increases in cold tolerance[15,19–21,43]. Given that GPCRs are expressed in diverse tissues, it is hypothesized that knockdown of GPCR genes may exhibit a tendency toward abnormal increases in cold tolerance.

In the temperature acclimatization assay, the *srh-40* mutant animals exhibited a decreased ADL temperature response and an increased survival rate. A previous study also showed that the *kqt-2* mutant lacking a KCNQ potassium channel exhibited reduced ADL temperature response and abnormal temperature acclimatization[22], which were equivalent to the phenotypes observed in the *srh-40* mutants. However, the *kqt-3* mutant lacking another KCNQ channel exhibited increases in ADL thermal responsiveness and survival rate in temperature acclimatization[22]. These data, including ours, clearly suggest that ADL activity influences temperature acclimatization. Nevertheless, it appears that complex mechanisms that yet remain unidentified exist beyond these controversies, which requires further research.

The ectopic expression of SRH-40 in the ASER neuron conferred warmth sensitivity on the non-warmth-sensing neuron ASER, and this acquired temperature sensitivity was interfered by the mutation of Gα (*goa-1*, *gpa-3*, and *egl-30*), GC (*gcy-5*), and CNGC (*tax-4*) (Fig. 3), suggesting that an SRH-40-dependent thermosensory signal is transduced by these molecules in ASER. Regarding the ADL neuron, the SRH-40-dependent thermosensory signal is transduced by Gα EGL-30 and TRPV channels (Fig. 2). These data suggest that CNGCs and TRPV channels are activated through the common Gα protein in distinct type of neurons. One such example has been reported in previous study showing olfactory receptor reprogramming[44]. An olfactory receptor, ODR-10, expressed in the AWA neuron is required for attraction to diacetyl, and the ectopic expression of ODR-10 in the AWB aversive neuron was found to induce avoidance to diacetyl[44]. In both AWA and AWB neurons, the Gα protein ODR-3 was commonly required for the transduction of ODR-10-mediated olfactory signaling, and the TRPV channels OSM-9/OCR-2 and CNGCs TAX-2/TAX-4 were required in the respective AWA and AWB neurons[44]. Hence, CNGCs and TRP channels are activated by the common Gα ODR-3 in the two different types of neurons. Therefore, a set of a receptor and Gα could drive multiple downstream signaling pathways depending on components expressed in different cellular contexts. Remarkably, in addition to Gα EGL-30, a mutation of another Gα *goa-1* interfered the SRH-40-mediated thermal signaling in ASER ectopically expressing SRH-40 (Fig. 3), raising a possibility that multiple types of Gα transfer SRH-40-mediated thermal signaling toward CNGCs in ASER.

We selected *Drosophila* S2R+ cells as the in vitro expression system for SRH-40 to evaluate its temperature responsiveness because the culture temperature was equivalent to that for *C. elegans*, and these cells could abundantly express SRH-40. The Ca²⁺ᵢ concentration in vector-transfected S2R+ cells increased upon heating, whereas the absence of extracellular Ca²⁺ reduced the heat-induced increase in Ca²⁺ᵢ concentration (Fig. 4). These findings indicate that S2R+ cells are intrinsically temperature-sensitive, and extracellular Ca²⁺ flows into cells upon heating through an unknown mechanism. Deep sequencing analysis has revealed that S2R+ cells scarcely express known temperature sensors such as TRP channels and ionotropic receptors (Irs)[45].

Although Ir25a and Ir93a might be expressed, which are co-receptor and involved in cooling and heating responses[46], another component Ir68a was not expressed in S2R+ cells[45], and there is no evidence indicating that a pair of Ir25a and Ir93a forms a functional heat sensor.

We confirmed an increase in $Ca^{2+}_i$ concentration in cells expressing OSM-9/OCR-2 by heat stimulation, which are known to be thermosensitive, suggesting that temperature response is measurable in S2R+ cells. Based on our in vivo study, we determined that SRH-40, $G_{\alpha q}$ EGL-30, and OSM-9/OCR-2 cooperatively function to respond to heating in ADL; however, cells expressing all these components did not show the additional increases in $Ca^{2+}_i$ concentration compared with cells expressing only OSM-9/OCR-2. It is probable that unidentified intermediate component(s) between SRH-40 and OSM-9/OCR-2 are missing in S2R+ cells or that SRH-40 and TRPVs function independently in temperature sensation in ADL. Irrespective of the mechanism, we propose that both SRH-40 and OSM-9/OCR-2 constitute a dual thermosensory signaling machinery in ADL.

The concentration of $Ca^{2+}_i$ in cells expressing SRH-40/TAX-2/TAX-4 significantly increased upon heating (Fig. 4). Importantly, there was no such increase in $Ca^{2+}_i$ concentration when cells expressed any other combinations, including SRH-40 alone, TAX-2/TAX-4, SRH-40/GOA-1, or GOA-1/TAX-2/TAX-4. These data suggest that SRH-40 and CNGCs are vital components for temperature response in S2R+ cells and are consistent with the evidence that ectopically expressing SRH-40 in warmth-insensitive ASER acquired thermal responsiveness through $G_{\alpha}$ and CNGCs, and that the ADL temperature response was driven by SRH-40. It is possible that SRH-40 expression induces the expression, stabilization, or localization of other thermosensitive molecules in S2R+ cells; however, it occurs only in combination with TAX-2/TAX-4.

The findings of this study provide a series of evidence that GPCRs act in temperature signaling in the thermosensory neuron, regulating temperature acclimatization in animals. Together with our previous data indicating that the TRPV channels OSM-9/OCR-2 are temperature-sensitive in the ADL neuron[5,19,22], we propose that SRH-40 and TRPV channels constitute a dual thermosensory signaling machinery in a single sensory neuron, functioning either in parallel or independently. Another GPCR, rhodopsin, has been reported to contribute to thermotaxis in Drosophila[6,8,47] and mammalian sperm[48], suggesting that the GPCR-mediated temperature sensation could be evolutionarily conserved. Our study may provide insights into clarifying the mechanisms underlying the temperature sensitivity of GPCRs across species.

## Methods

### C. elegans strains
The N2 (Bristol) strain was used as the wild-type C. elegans in all experiments. The strains were maintained according to standard procedures. C. elegans were maintained on nematode growth medium (NGM) containing 2% (w/v) agar and a bacterial lawn of Escherichia coli OP50 in 35- or 60-mm dishes at 15 °C or 20 °C and transferred to new NGM every 5 or 3 days. This adult hermaphrodites C. elegans was used in the whole assay. Male C. elegans were utilized for mating with hermaphrodites to generate multiple mutants. We used the C. elegans strains as shown in Supplementary Table 2.

### Cold tolerance and temperature acclimatization assays
We used these protocols: [15 °C → 2 °C (48 or 72 h)], [20 °C → 2 °C (5–7 or 48 h)], and [15 °C → 25 °C → 2 °C]. One or two well-fed adult animals were placed on a 35-mm dish containing NGM for egg-laying. The adult animals were removed after one night, and the progeny were cultured for 120–150 h at 15 °C, or 85–90 h at 20 °C. Plates with approximately 100 animals were transferred to 2 °C after being acclimatized at designated optimal conditions (e.g., 25 °C, 0, 3, or 5 h). After 48 h at 2 °C, the plates were transferred and incubated at 15 °C, and the number of dead or alive animals was calculated on the next day. As an exception, the time for cold stimulation at 2 °C was 72 h and 5–7 h in

Supplementary Fig. 3d, e, respectively, to observe the mutant phenotype under conditions where the survival rate of the wild-type is approximately 50%. Cold tolerance and temperature acclimatization assays were conducted as described previously[5,15,20,22,43,49].

### RNA interference of GPCR genes by feeding RNAi
RNAi screening was performed using the "feeding RNAi" method as described previously using the RNAi library constructed by Julie Ahringer's group[50]. This screening was conducted at 20 °C on NGM plates supplemented with 1 mM IPTG (Wako, Osaka, Japan) and 25 µg/mL carbenicillin (Wako). RNAi food, an HT115 bacterial strain expressing dsRNA against a target gene, was cultured at 37 °C with shaking for 5 h in LB medium consisting of 100 mg/mL ampicillin. The pelleted bacteria dissolved in 75 µL LB were spread onto an NGM plate and used as an RNAi plate. The next day, L4 larva ~ adult $P_0$ animals were placed on the RNAi plate and cultivated overnight at 20 °C; then, these animals were transferred onto another RNAi plate to lay eggs overnight. $P_0$ adult animals were removed from the RNAi plate, and after 3 days, a cold tolerance assay [20 °C → 2 °C] was performed.

### Confocal microscopy
To prepare samples for confocal microscopy, 2% (w/v) agarose gel on a glass microslide was covered with 10 µL drops of 100 mM NaN₃, after which 4–6 worms were placed and covered with a cover slip. Fluorescence images were acquired by confocal laser microscopy (FV1000-IX81 with GaAsP PMT, Olympus Corporation, Tokyo, Japan) using the FV10-ASW (Ver. 04.02) software (Olympus).

### DiI staining (dye-filling with DiI)
DiI (1,1'-dioctadecyl-3,3,3',3'-tetramethylindocarbocyanine perchlorate) staining was performed essentially according to the protocol in Anatomical Methods in WORMATLAS (www.wormatlas.org/EMmethods/DiIDiO.htm). DiI is a red fluorescence dye used to visualize amphid neurons (ADL, ASH, ASI, ASJ, ASK, and AWB), inner labial neurons (IL1 and IL2), and phasmid neurons (PHA and PHB). A 1 mL tube containing well-fed worms was incubated at room temperature for 1 h with 200 µL M9 buffer containing 1 µg/mL DiI dye. The stained worms were washed two times with 1 mL M9 buffer and transferred onto 35-mm NGM plates containing 2% (w/v) agar and a bacterial lawn to distain waste. After cultivating at 20 °C for approximately 12 h, the stained worms were observed under a confocal microscope.

### ADL-specific knockdown of egl-30
RNAi knockdown in specific neurons was performed by transgenically introducing RNAi as described previously[51]. The gene sequences for RNAi were amplified according to the RNAi library constructed by Julie Ahringer's group[50]. The ADL-specific promoter (srh-220 promoter) was fused with egl-30 fragments in cloned sense and antisense directions.

### CRISPR/Cas9
Knockout mutants were developed using a co-CRISPR protocol using the dpy-10 marker[52]. We engineered nonsense mutations in addition to a restriction site in the first half of GPCR genes or deletion in GPCR genes. Guide RNAs (gRNAs) containing "GG" DNA sequences at the 3' end (3'GG) were used[53], which were expressed by the PU6 promoter. pDD162 (Peft-3::Cas9) was injected into N2 animals at 50 ng/µL with the targeted gRNA and dpy-10 gRNA in plasmid vectors at 25 ng/µL, the targeted donor oligonucleotide, and the dpy-10(cn64) donor oligonucleotide at 500 nM. As the dpy-10(cn64) mutant exhibits a roller phenotype, we picked Roller animals at $F_1$ and screened for the mutation in the targeted gene by PCR and digestion with restriction enzymes. After detecting the mutation in the targeted gene, the roller mutant was crossed with wild-type worms. All mutations were verified by direct sequencing to confirm the desired mutations at the final step.

Mutations, plasmids, and primers used for CRISPR/Cas9 are listed in Supplementary Tables 3, 4 and Supplementary Data 2.

## Molecular biology

All the promoters of GPCR genes used for GFP expression analysis were generated by PCR from *C. elegans* genomic DNA. These promoters contained 0.5–4 kb upstream of the predicted translational start site, including part of the first exon of a given gene product. The genomic fragment was inserted into the GFP vector pPD95.75, followed by the *unc-54* untranslated region. pMIU66 (*srh-40 full length::gfp::srh-40 3'UTR*) was generated by PCR; an *srh-40* full-length genomic fragment without stop codon and the *srh-40* 3'UTR were amplified by PCR from *C. elegans* genomic DNA, and then *gfp* was fused to the 3' end of *srh-40*. Expression analysis plasmids were injected at 80–200 ng/µL. pMIU28 (*srh-220p::srh-40 cDNA*) contained 2462 bp upstream of the promoter sequence of *srh-220* and *srh-40* cDNA amplified by PCR from the cDNA library. pMIU29 (*srh-40p::srh-40 cDNA*) contained 394 bp upstream of the promoter sequence of *srh-40* gene and *srh-40* cDNA. Rescue plasmids were injected at 5 ng/µL. pMIU30 (*gcy-5p::srh-40 cDNA*) contained 1998 bp upstream of the promoter sequence of *gcy-5* and *srh-40* cDNA. pKOH244 (*srh-220p::egl-30(Q205L) cDNA*) contained *egl-30(Q205L)* cDNA downstream of the *srh-220* promoter, and pKOH244 was injected at 10 ng/µL. pKOH252 (*srh-220p::egl-30 RNAi antisense*) and pKOH253 (*srh-220p::egl-30* RNAi sense) contained a part of *egl-30* from the genome (5'-TTCCAGTC-TATG…GAGACACCACCA-3') for feeding RNAi downstream of the *srh-220* promoter. pKOH252 contained an *egl-30* fragment amplified by PCR from 3' to 5' (GTTTAGGTGGTG… CATAGACTGGAA), and pKOH253 contained an *egl-30* fragment amplified by PCR from 5' to 3' (TTCCAGTCTATG…GAGACACCACCA). pKOH252 and pKOH253 were injected at 50 ng/µL. pRF4 *rol-6(gf)* at 30–40 ng/µL was used as a transgenic marker for expression analysis or Ca$^{2+}$ imaging in the ADL of *C. elegans*. pKDK66 (*ges-1p::NLS::gfp*) and pAK62 (*AIYp::gfp*) at 30 and 50 ng/µL, respectively, were used as transgenic markers for temperature acclimatization testing of transgenic strains. pMIU156 (*sre-1p::yc4.12*) contained *yc4.12*[38], which was generated by mutating to the calcium-binding domain of yc2.12, downstream of the *srh-220* promoter. YC4.12 was kindly provided by Dr. Mori. A pAc5.1-V5-His (pAc5.1) containing *Drosophila* actin 5 C or a pMT vector containing *Drosophila* metallothionein gene promoter was used for introducing cDNA to S2R+ cells. pMIU97 (*pMT-srh-40* cDNA) contained *srh-40* cDNA in the pMT vector. pMIU133 (*pMT-egl-30* cDNA*::T2A::mCherry*) contained *egl-30* cDNA without stop codon and the DNA sequence of the T2A peptide (5'-GGATCAGGAGAAGGAAGAGGATCACTTCTTACAT GTGGAGATGTTGAAGAAAATCCAGGACCA-3') for simultaneously expressing the two genes and *mCherry* in the pMT vector. pMIU136 (*pAc5.1-osm-9* cDNA*::T2A::ocr-2* cDNA) contained *osm-9* cDNA without a stop codon, T2A sequence, and *ocr-2* cDNA in the pAc5.1 vector. pKOH297 (*pAc5.1-tax-2* cDNA*::T2A::tax-4* cDNA) contained *tax-2* cDNA without stop codon, T2A sequence, and *tax-4* cDNA in the pAc5.1 vector. pMIU146 (*pAc5.1-goa-1* cDNA*::T2A::mCherry*) contained *goa-1* cDNA without stop codon, T2A sequence, and *mCherry* in the pAc5.1 vector. pKOH233 (*pAc5.1-mCherry*) contained *mCherry* in the pAc5.1 vector. Plasmids and primers constructing in this study are listed in Supplementary Table 4 and Supplementary Data 2.

## Ca$^{2+}$ imaging in *C. elegans*

Animals expressing yellow cameleon 3.60 driven by the *sre-1* promoter *sre1p::yc3.60* (pTOM63) were used for Ca$^{2+}$ imaging in ADL. Animals simultaneously expressing *C. elegans* codon-optimized *GCaMP8s (sensitive)* driven by the *flp-6* promoter pMIU34 (*flp-6p::Ce-GCaMP8*) and pKOB006 (*gcy-5p::tagRFP*) were used for Ca$^{2+}$ imaging in ASER. Animals were cultivated at 15 °C, expect for the experiment of Fig. 2i and Supplementary Fig. 3j in which were cultivated at 20 °C. Adult animals were attached to a 2% (w/v) agar pad on glass, immersed in M9

buffer, and then covered with glass. The M9 buffer contained 3 g KH$_2$PO$_4$, 6 g Na$_2$HPO$_4$, 5 g NaCl, and 1 mL of 1 M MgSO$_4$ in 1 L H$_2$O. Samples were placed on an ITO glass-based thermocontroller (Tokai Hit Co., Fujinomiya, Japan) mounted on the stage of an Olympus IX81 or BX61 microscope (Olympus Corporation, Tokyo, Japan) at the initial imaging temperature for a few minutes. Fluorescence images of donor and acceptor fluorescent proteins in yellow cameleon, or green and red fluorescence in GCaMP8 and tagRFP, were simultaneously captured using the following optical system: an EVOLVE512 EM-CCD camera (Photometrics, Tucson, AZ) with dual-view (Molecular Devices, San Jose, CA) as used in Figs. 2i and 3e, f; iXon Ultra 888 EM-CCD camera (Oxford Instruments, Abingdon, UK) with a split-view model of CSU-W1 (Yokogawa Electric Corporation, Tokyo, Japan) as used in Figs. 1l and 3a–c; ORCA fusion BT (Hamamatsu Photonics, Hamamatsu, Japan) with dual-view as used in Figs. 1m, 2f, h and 3d; EVOLVE512 EM-CCD camera with a W-View 2 C gemini (Hamamatsu Photonics) as used in Figs. 2e and 3f; and ORCA fusion with a W-View 2 C gemini as used in Figs. 2d, e, 3e, f, and Supplementary Fig. 3j. Images were captured with a 10–60 ms exposure time with 1 × 1 binning (Figs. 1l, m, 2d, e, i, and 3a–c, e, f, and Supplementary Fig. 3j) or 2 × 2 binning (Figs. 2f, h and 3d). The temperature on the agar pad was measured using a thermometer (TP-OT-B14M, Tokai Hit Co., Ltd.). For each imaging experiment, the fluorescence intensity was measured using the Meta-Morph (Molecular Devices) image analysis system. Relative changes in intracellular Ca$^{2+}$ concentrations were measured as the change in the acceptor/donor fluorescence ratio of YC3.60 protein or in the green/red fluorescence ratio of GCaMP8 and tagRFP. We used YC3.60 for Ca$^{2+}$ imaging of ADL and GCaMP8 or ectopic expression analysis in ASER, according to previous studies[5,13,15,19,20]. Bar graphs for Ca$^{2+}$ imaging of ADL indicate the averaged YFP/CFP ratio between 211–220 s in Figs. 1l, 2d–f, h, i and Supplementary Fig. 3j, or 101–110 s in Fig. 1m, where the wild-type animals showed the maximum changes. When we conducted the analyses for Fig. 2d in winter, the ADL temperature responses of wild-type animals increased compared with results shown in Fig. 2e, wherein the analyses were conducted in winter to spring seasons; hence, this observed difference may have been caused by humidity and other unknown factors, as described previously[21,49]. In vivo Ca$^{2+}$ imaging in *C. elegans* was conducted according to previous studies[15,24,49,54].

## Stable S2R+ cell line expressing SRH-40

To establish a *Drosophila* Schneider 2 R+ (S2R + ) cell line harboring an inducible SRH-40 gene, cells were transfected with 1 µg of pMT-*srh-40* and 0.05 µg of pBS-PURO using the XtremeGENE9 DNA transfection reagent (Roche, Basel, Switzerland) dissolved in 1× OPTI-MEM medium (Gibco, Grand Island, NY). After incubation for 48 h, the transfected cells were washed two times with Schneider's *Drosophila* medium and further cultured at 25 °C. After 2 days, the cells were reseeded on a 100-mm dish and cultured in the presence of 10 µg/mL puromycin. The culture medium containing puromycin was replaced every 4 days. After 2 weeks, the dish was washed with medium, and the surviving cells were collected as polyclonal cell lines. *Drosophila* S2R+ cell were obtained from the Drosophila Genomics Resource Center (DGRC Stock 150; https://dgrc.bio.indiana.edu//stock/150; RRID:CVCL_Z831).

## Quantitative PCR

L4 larva - adult *eri-1; lin-15B* animals were cultivated on the RNAi plate overnight at 20 °C; then, these animals were transferred onto another RNAi plate to lay eggs overnight. The progeny in which the *srh-40* gene was knocked down were cultured for 85–90 h at 20 °C. The *srh-40* S2R + cells were treated with 500 mM CuSO$_4$ (Wako) and cultured at 25 °C for 48 h. Total RNA was extracted from the cells using the Maxwell RSC simplyRNA Cells Kit, according to the manufacturer's instructions (Promega, Madison, MI). cDNA was reverse-transcribed from total RNA (250 ng) using the iScript Advanced cDNA Synthesis kit for RT-PCR,

according to the manufacturer's instructions (BioRad, Hercules, CA). Quantitative PCR was performed using each primer, Sso Fast EvaGreen Supermix (Thermo-Fisher), and the CFX96 real-time PCR detection system (BioRad). Comparative gene expression was analyzed using the 2^-ΔΔCq methods and calculated based on *eri-1; lin-15B* or *srh-40* S2R+ cells not treated with $CuSO_4$, used as a control. Tubulin alpha chain *tba-1* in *C. elegans* or *Ribosomal protein 49* (*Rp49*) in *D. melanogaster* was used for normalization. The quantitative PCR was conducted in duplicate or triplicate. The following primer sets were used: *tba-1* Fwd, AGACCAACAAGCCGATGGAG; *Rp49* Rev, GCATCTTCCTTTCCGGT-GAT; *Rp49* Fwd, CCGCTTCAAGGGACAGTATCTG; *Rp49* Rev, CACGT TGTGCACCAGGAACTT; *srh-40* Fwd, CCATACAGAAAGTTCACTTTGG; *srh-40* Rev, TGTATTCGACCTGGCAAC.

## Ca²⁺ imaging in *Drosophila* S2R+ cells

The *srh-40* S2R+ cells were cultured in 60-mm dishes (Falcon, AZ) at 25 °C in 3 mL Schneider's *Drosophila* medium (Gibco), supplemented with 10% heat-inactivated fetal bovine serum (Gibco) and penicillin–streptomycin (50 mg/mL and 50 units/mL, respectively, Gibco). Puromycin (10 μg/mL) was added for maintaining *srh-40*-positive cells. For transient transfection, S2R+ cells were seeded on 12-mm round cover glasses (Matsunami, Tokyo, Japan) coated with poly-L-lysine in 35-mm dishes, and a mixture of 1.2 μg of the genes and *mCherry* marker in *pAc5.1* or *pMT* vector was transfected using OPTI-MEM medium (1×, Gibco) containing X-tremeGENE9 DNA transfection reagent (Roche). A DNA mixture transfected in the SRH-40 stable cell line shown in Fig. 4 and Supplementary Fig. 4b, c consists of the following compositions: "OSM-9, OCR-2" and "SRH-40, OSM-9, OCR-2" containing 0.5 μg *osm-9::T2A::ocr-2* in *pAc5.1* vector, 0.5 μg *pAc5.1* vector, and 0.2 μg *pAc5.1-mCherry*; "SRH-40, EGL-30, OSM-9, OCR-2" containing 0.5 μg *osm-9::T2A::ocr-2* in *pAc5.1* vector, 0.5 μg *egl-30::T2A::mCherry* in *pMT* vector, and 0.2 μg *pAc5.1* vector; "Control" and "SRH-40" containing 1.0 μg *pAc5.1* vector and 0.2 μg *pAc5.1-mCherry*; "TAX-2, TAX-4" and "SRH-40, TAX-2, TAX-4" containing 0.5 μg *tax-2::T2A::tax-4* in *pAc5.1* vector, 0.5 μg *pAc5.1* vector, and 0.2 μg *pAc5.1-mCherry*; "GOA-1, TAX-2, TAX-4" and "SRH-40, GOA-1, TAX-2, TAX-4" containing 0.5 μg *goa-1* in *pAc5.1* vector, 0.5 μg *tax-2::T2A::tax-4* in *pAc5.1* vector, and 0.2 μg *pAc5.1-mCherry*. "SRH-40, GOA-1" containing 0.5 μg *goa-1* in *pAc5.1* vector, 0.5 μg *pAc5.1* vector and 0.2 μg *pAc5.1-mCherry*.

To induce the expression of SRH-40 fused to the metallothionein promoter pMT in the polyclonal stable cell line, 500 μM $CuSO_4$ (Wako) was added to the cell culture during transfection. The transfected/induced cells were cultured for 40–70 h before experiments. Ca²⁺ imaging was performed at room temperature; warming stimuli were provided by perfusion of heated bath solution, and cooling sti-muli were induced by perfusion of an ice-chilled bath solution. Temperature was monitored using a TC-344B temperature controller (Warner Instruments, Holliston, MA); the temperature probe was located just beside the cells. We applied successive stimuli, a cold stimulation (approximately 15 °C) followed by a heat stimulation (approximately 35 °C) (Fig. 4a–g, Supplementary Fig. 4b, c) or a heat stimulation (approximately 35 °C) (Fig. 4h–k) and repeated them twice. The Ca²⁺ concentrations were measured using Fura-2; Fura-2 was excited at 340/380 nm, with emission being captured at 510 nm. A cover glass containing the transfected/induced S2R+ cells was incu-bated at 25 °C for more than 1 h with 1 mL culture media containing 5 μM Fura-2/AM (F-1201, Life Technologies, Carlsbad, CA), 0.02% pluronic F-127 detergent (P2443, Sigma-Aldrich, St. Louis, MO), and 1 mM probenecid (162-26112, Wako). Fluorescence was captured using a CCD camera with 4 × 4 binning, CoolSNAP DYNO (Photo-metrics, Netherlands). Data were measured and analyzed using the Nikon NIS element AR (Nikon Corporation, Tokyo, Japan). The external Ca²⁺ (+) bath solution contained 130 mM NaCl, 5 mM KCl, 2 mM $MgCl_2$, 2 mM $CaCl_2$, 10 mM TES, and 30 mM sucrose at pH 7.2,

adjusted using NaOH. For Ca²⁺-free experiments, 2 mM $CaCl_2$ was replaced by 5 mM EGTA, pH 8.0; 500 μM probenecid was dissolved in all bath solutions before conducting the experiments to retain Fura-2 in the cells.

## Measurement of fluorescence intensity in ADL neurons

Worms were immobilized and mounted with 100 mM $NaN_3$ on agar pads on glass microscope slides. We investigated whether the expression levels of *srh-40* in ADL depended on the cultivation tem-perature. Wild-type animals expressing *srh-40p::GFP* at various culti-vation temperature conditions were used in this study. Worms were imaged using an Olympus IX81 microscope (Olympus). Fluorescence was observed using the EM-CCD camera EVOLVE512 (Photometrics). Excitation light was generated by a SPECTRA4 Light Engine (LUMEN-COR). Fluorescence was acquired with 200 ms exposure times, 52% (133/255) light power of 475 nm LED in SPECTRA4 Light Engine, 70 EM gain of EM-CCD camera, 40× (objective lens), and 1 × 1 binning (1 pixel = 0.4 μm).

## 1-Octanol avoidance assay

Animals were grown at 20 °C for all behavioral assays. The avoidance assay against aversive odorants using 1-octanol was performed as described previously[44,55,56]. Adult animals were briefly washed off with NG buffer from the cultivation plates, collected, and placed on the center of a 90-mm chemotaxis assay plate. The animals were counted after 12 min. The avoidance index was calculated according to ref. 56. The animals remaining in the center of the plate were omitted from scoring.

## Pheromone avoidance assay

Crude pheromone for the avoidance assay was provided by Dr. T. Kawano and K. Sakamoto. To obtain this pheromone, *C. elegans* were incubated in 500 mL liquid medium at 20 °C for 72 h. The worms were allowed to settle and precipitate at low temperature. The nematode wet volume was 5 mL. The supernatant was collected and centrifuged, and the remaining food *E. coli* was separated, after which the super-natant was further collected. The supernatant was treated by activated carbon chromatography, washed with distilled water, eluted with methanol, and concentrated, and then the methanol was volatilized and dissolved in 5 mL distilled water. From this solution, 5 μL was used for the one droplet assay.

Animals were grown at 20 °C in the absence of food. Adult animals were briefly washed off with M9 buffer from the plates cultivated at 20 °C for 3 days and collected, and 20–30 worms were placed on non-seeded 60-mm NGM plates. After 6 h, approximately 5 μL of crude pheromone dissolved in water was delivered near to the anterior region of an individual animal using glass capillaries, and its backward movements were observed. The fraction reversing was calculated by dividing the number of backward responses by the total number of trials. The avoidance assay against the aversive chemical was con-ducted based on previous studies[34,35,57].

## Statistical analysis

All data are expressed as mean ± SEM. Statistical tests were conducted using Mac statistical analysis version 3 (Esumi, Japan). We assumed that datasets in the cold tolerance assay, temperature acclimatization assay, and Ca²⁺ imaging for ADL neurons shown in Figs. 1e–m, 2, Supplemen-tary Figs. 2a–c, and 3d–f, h, j followed a normal distribution according to previous experiments using the same experimental procedures[43]. The sample sizes in dataset for cold tolerance assay, temperature acclima-tization assay, and Ca²⁺ imaging in *C. elegans* shown in Figs. 1–3, Sup-plementary Figs. 2, 3 determined based on previous experiments using the same experimental procedures: at least 9 independent experiments were performed in these experiments[13,43]. For Ca²⁺ imaging in *Droso-phila* S2R+ cells in Fig. 4, Supplementary Fig. 4b, c, two to five

independent trials were repeated three or more times on separate days. The number of biological replicates is described in the figure legend. Single comparisons with an equal variance and normal distribution were performed using unpaired Student's *t*-test. Single comparison of the data shown in Fig. 4k was performed using the Mann–Whitney *U* test. Multiple comparisons with normal distribution were conducted using one-way ANOVA with Dunnett's test or Tukey–Kramer's test. If any group in the data was not normally distributed or had a small sample size, multiple comparisons were conducted using Kruskal–Wallis's test with the Steel test or Steel–Dwass's test. Not significant (n.s.), Single asterisk (*), and double asterisks (**) indicate $P \geq 0.05$, $P < 0.05$, and $P < 0.01$, respectively.

## Reporting summary
Further information on research design is available in the Nature Portfolio Reporting Summary linked to this article.

## Data availability
The data generated during in this study are available within this article and its supplementary information or from the corresponding author upon request. Source data are provided with this paper. Source data includes raw data and statistical analysis data. This study makes use of the publicly available databases "CeNGEN (https://www.cengen.org)", which is gene expression profiles of every neuron in *C. elegans*". Source data are provided with this paper.

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

## Acknowledgements

We thank T. Ujisawa for technical advice; I. Mori, K. D. Kimura, S. Ihara, Y. Iino, S. Mitani, and K. Sakamoto, T. Kawano for sharing samples; Some strains were provided by the National Bioresource Project (Japan), and the *Caenorhabditis* Genetic Center CGC, which is funded by NIH Office of Research Infrastructure Programs (P40 OD010440); and members of the Kuhara Laboratory for comments and discussions. Funding: JSPS KAKENHI (23K14235) (K.O.). The Naito Foundation, Takeda Science Foundation, Hirao Taro Foundation of Konan Gakuen for Academic Research, and the Asahi Glass Foundation, Research Foundation for Opto-science and Technology, Shimadzu Science Foundation, Brain Science Foundation, and Toray Science Foundation (A.K. and A.O.). AMED PRIME (23gm6510004h0003), G7 Foundation, and JSPS KAKENHI (21H02534, 21K19279, and 22H05512) (A.K.). The Cooperative Study Program (21-222, 23NIPS112) of National Institute for Physiological Sci-ence, Joint Research of the ExCELLS (22EXC333), and KAKENHI Thermal Biology from MEXT Japan (15H05928) (M.T. and A.K.). JSPS KAKENHI (21K06275) (A.O.). JSPS KAKENHI (21H02531), the AMED PRIME (23gm6510014h0002), and Takeda Science Foundation (T.S.).

## Author contributions

K.O., T.S., M.T., T.M., A.O., and A.K. designed and performed the experiments, interpreted the results, and wrote the final report.

## Competing interests

The authors declare no competing interests.

### Ethical approval

All animal treatments in this research were performed in accordance with the Japanese Act on Welfare and Management of Animals (Act No. 105 of October 1, 1973; latest revisions Act No. 51 of June 2, 2017, Effective June 1, 2018). All experimental protocols were approved by the Institutional Animal Care and Use Committees of Konan University.
