## [Peer Review File · Nature Communications]

REVIEWER COMMENTS

Reviewer #1 (Remarks to the Author):

In this article, Ohnishi and collaborators present a neurogenetic study in which they identified a GPCR expressed in the ADL neurons of *C. elegans* as a potential molecular thermo-sensor mediating temperature acclimatization to enhance cold tolerance. The latter phenomena is a very interesting physiological plasticity case, in which worms better survive a cold treatment (2°C) if they were pre-treated for some time at a cool temperature (15°C) rather than at warmer temperature (25°C). In previous studies, the group of Dr. Kuhara showed that ADL sensory neurons mediate the thermal acclimatization and that two TRPV channels, OSM-9 and OCR-2, expressed in ADL neurons might contribute to temperature detection relevant for cold tolerance. However, their data suggested that TRPV channels might be accessory and that additional thermosensory molecules were involved. In order to identify these unknown thermo-responsive molecules, the author hypothesized that it could be a GPCR and conducted a RNAi screen covering most GPCR encoded in the worm genome. They identified many GPCRs whose knockdown could either up-regulate or down-regulate cold tolerance. Knocking down *srh-40* with RNAi or with CRISPR-engineered mutations had a strong impact on cold tolerance. They show that SRH-40 is (i) expressed in ADL, (ii) required for ADL to develop normal heat-evoked calcium transients in response to minute-lasting heat stimuli from 13 to 27°C, (iii) able to confer thermal sensitivity to another *C. elegans* thermosensory neuron (ASER) and (iv) able to enhance the pre-existing thermal responsiveness in *Drosophila* S2R+ cells.

The article is well written overall and easy to follow, albeit it only cursorily discusses the results. The RNAi screen is impressive and a great achievement on its own. The data showing the requirement for *srh-40* and the presumable locus of action of SRH-40 in ADL, as well as its role for ADL calcium activity are convincing. In contrast, the dissection of the downstream mechanisms is somewhat confusing. The result of SRH-40 ectopic expression experiments are compatible with the model in which this GPCR is a thermosensory component, but other explanations exist and the data also raise some important questions/doubts. Furthermore, the connection between GPCR and TRPV channel is not addressed appropriately, despite being a very important aspect of the story (and this both regarding the experiments conducted and the discussion).

While the presented findings are very interesting and important in the field, some evidence/controls are lacking on some important aspects and I would request that the authors fully address the concerns below, before I can recommend the article for publication.

Major points:

1) Fig. 1f, g,h: It is not clear why the rescue experiments are conducted after 7h at 25°C, while 3h and 5h are used elsewhere. The entire dataset at 3, 5 and 7 h is needed for the reader to evaluate the temporal window for which *srh-40* is relevant in ADL.

2) In order to interpret the ASER ectopic expression experiments, we need to have one-to-one comparisons with ADL. We therefore need to see ASER calcium activity in *egl-3* mutant and ADL calcium activity in *goa-1* mutant. The justification brought by the authors to focus on EGL-30 because it could regulate TRPV channel (P3 L35) is hard to follow, because we have no solid reason to believe SRH-40 must work upstream of TRPV channels (neither prior nor after the present study). Of note, *goa-1* and *egl-30* are expressed in both ADL and ASER, based on CENGEN data.

3) The connection between TRPV and SRH-40-dependent thermo-responsiveness in ADL is unclear. Data in Fig. 2G, suggest *srh-40* and *ocr-2* might work in the same genetic pathway in regulating ADL warmth-evoked response. But data are not conclusive because of a potential floor effect in the measure, and because *osm-9* was surprisingly left aside of the analysis. Expression in S2R+ cells, suggest that SRH-40 has no cumulative effect with OSM-9 OCR-2 in this very artificial system (comparing values in Fig.4 and corresponding expanded data). The following experiments are needed:

a) ADL calcium imaging in triple *ocr-2 osm-9; srh-40* mutants (ideally the quadruple mutant *ocr-2 osm-9; ocr-1 srh-40*)

b) If response is still there in the triple, control imaging with a calcium-defective version of cameleon, to demonstrate that residual temperature-evoked response is not an artifact of the sensor.

c) Survival in *ocr-2 osm-9; srh-40* mutants, compared to *osm-9 ocr-2* double and *srh-40*.

In addition, the authors should explicitly discuss what they believe the connection is between TRPV channels and SRH-40 GPCR (notably in the light of the proposed additional experiment results). Early in the manuscript they state that GPCR might modulate TRP channel (notably via EGL-30). Later it looks like it could be two separate pathways. This is very confusing.

4) What is the cause of the temperature-evoked up-steps seen in S2R+ cells in 'Control' situation? I note that it is partially decreased by Ca²⁺ washout, which indicates it is not (or at least not entirely) the result of fura-2 being influenced by temperature. Should we conclude that S2R+ cells have already some quite marked intrinsic thermal responsiveness? This must be discussed. In addition, the statement on P5 L4, must be changed from "was evoked by SRH-40" to "was enhanced by SRH-40".

5) The Control situation in Fig. 4 is not well explained. How could there be only one control dataset, when the control is supposed to be a transfected but un-induced cell preparation? There must be one control per transfected plasmid mix. The total amount of DNA matters a lot for expression in transfection experiments, therefore, proper controls are needed with matching DNA content.

6) Generally speaking, it is possible that SRH-40 does not display temperature-dependent G-protein activation activity, but is simply required for the expression, the stabilization or the localization of another thermo-dependent molecule. This must be stated very explicitly and the claims toned down in the entire manuscript.

Minor points

1) Fig. 1d brings little in its current form. I guess it is ranked by alphabetical order of the RNAi targets, which is arbitrary and has no helpful meaning. I would suggest to rank the RNAi targets based on the

survival rate and highlight *eri-1*; *lin-15B* with an arrow. This will provide a more useful graphic that will not only look nicer but, more importantly, also better present the survival score distribution above and below that of the control.

2) P1 L22: “Exhaustive knockdown” should be rephrased (e.g., with “Systematic knockdown”) because not every GPCRs have been targeted.

3) P2 L27: “thermos-receptor” need to be replaced by “thermo-receptor”

4) Acknowledgements of the CGC should be complemented to follow their guidelines :
<https://cgc.umn.edu/acknowledging-the-cgc>

Reviewer #2 (Remarks to the Author):

In this manuscript, the authors (Ohnishi et al.) performed a targeted RNAi screen to find genes that affect cold tolerance in *C. elegans*. From more than one thousand GPCR genes, they found knock-down of *srh-40*, which is highly expressed in the ADL sensory neuron, can increase worms’ cold tolerance. Through calcium imaging experiments, they propose *srh-40* as a GPCR-type temperature sensor.

Overall, I think the authors did a lot of nice work, especially on the screen. The *C. elegans* genome contains a very large number of GPCRs. The characterization of the localization of many GPCRs (Extended Data Figure 1) is also very useful to many readers. The gene they found, *srh-40*, is a novel GPCR gene that is only expressed in ADL (CeNGEN). Its knock-down effect on cold tolerance is obvious and quite interesting.

However, I don’t think the data supporting *srh-40* as a temperature sensor is strong enough. First of all, the transition from the finding that *srh-40* affects cold tolerance to the hypothesis that SRH-40 is a temperature sensor is abrupt. I understand that SRH-40 is on the membrane and therefore could be suited for the role of a temperature sensor. But GPCRs in neurons can affect cold tolerance in the entire animal in many many ways, not necessarily by acting as a thermosensor. Secondly, in all the calcium imaging experiments, the calcium changes due to SRH-40 are quite mild, although significant. This makes it somewhat hard to conclude that SRH-40 is a bona fide thermosensor.

I also don’t understand what the authors try to propose as the signaling pathways downstream from *srh-40*. From the very beginning, the authors mentioned that “OCR-2/OSM-9 may act as accessorial warm receptor downstream of unknown thermos-receptor”. They also mentioned that “TRP channels are regulated by Gαq signaling”. These statements, fig. 1c, together with the Gαq/*egl-30* found to be

involved, all indicate that *srh-40* is the thermosensor upstream from OCR-2/OSM-9 in ADL. However, when authors ectopically express SRH-40 in ASER or *Drosophila* S2 cells, the downstream ion channels are TAX-2/TAX-4 in both cases (ASER only expresses TAX-2/TAX-4, but not the OCR-2/OSM-9 channel). The only explanation I can think of is that *srh-40* can activate both TAX-2/TAX-4 and OCR-2/OSM-9 channels in different contexts. These two channels are very different in nature. I have not known of any upstream player who can activate both. If the authors think *srh-40* or Gαq can activate both channels or know a precedent, it should be discussed.

The reason to express “SRH-40, TAX-2 and TAX-4 with or without GOA-1 in S2R+ cells”, but not OCR-2/OSM-9 is that “temperature response of TRPV channels in ADL (OCR-2/OSM-9) were detectable in S2R+”. This reasoning seems problematic to me. The authors propose that SRH-40 and OCR-2/OSM-9 are “dual thermo-sensing machinery”, and the results in ADL indicate that temperature responses from SRH-40 and OCR-2/OSM-9 are additive in ADL. Then, detecting temperature response from OCR-2/OSM-9 in S2R+ cells is not a problem, as long as SRH-40 plus OCR-2/OSM-9 has an increased response than OCR-2/OSM-9 alone, but the results failed to show this increase. The temperature-induced calcium response from OCR-2/OSM-9 is not saturated since adding ionomycin further increased intracellular calcium.

There are other places that, I think, need attention, or that I don't quite understand:

1.1. The cold acclimatization assay that has been performed throughout the manuscript is -- [25°C→15°C→2°C]. During the screen, *srh-40* was found to affect cold tolerance in another protocol -- [20°C→2°C]. This seems to suggest that *srh-40* is playing a role in cold tolerance per se, not necessarily cold acclimatization. This discrepancy has two potential problems. One is whether the mechanism investigated is about intrinsic cold tolerance or cold acclimatization. The other is that the authors probably should do calcium imaging to find out *srh-40*'s role during cooling since the protocol [20°C→2°C] does not involve a warming step. Many genuine thermosensors can only respond to a narrow temperature range. If there is a reason only to look at the response to warming, it should be stated.

1.2. The authors tested that the *srh-40* null allele did not affect worms chemotaxis to 1-octanol. Given that ADL's most well-known role is to sense pheromones, I think testing chemotaxis to pheromones, such as C3, could be a more appropriate control.

1.3. Whether a decreased calcium response in ADL (as seen in *srh-40* mutants) upon warming in general correlates with a better cold tolerance level. If so, how is it achieved (maybe out of scope)? Why not the opposite (increased calcium response correlates with worse cold tolerance)?

Some minor issues:

2.1. In fig. 1d, the X axis is better organized according to the order of its survival rate. The control is somewhere in the middle. Then, use different colors to highlight the control and *srh-40*. This gives the readers a sense of where *srh-40* is placed on the entire screen.

2.2. On page 5, line 2, the statement “the Ca²⁺i increase disappeared” is not accurate. I think the calcium increase was dampened, but it did not disappear.

2.3. There is no discussion. The last paragraph merely summarized their findings.

I think the manuscript has a very good start, and finding that SRH-40 affecting cold tolerance is very cool. I also think the mechanism by which SRH-40 affects cold tolerance should be looked into more carefully.

Reviewer #1 (Remarks to the Author):

In this article, Ohnishi and collaborators present a neurogenetic study in which they identified a GPCR expressed in the ADL neurons of *C. elegans* as a potential molecular thermo-sensor mediating temperature acclimatization to enhance cold tolerance. The latter phenomena is a very interesting physiological plasticity case, in which worms better survive a cold treatment (2°C) if they were pre-treated for some time at a cool temperature (15°C) rather than at warmer temperature (25°C). In previous studies, the group of Dr. Kuhara showed that ADL sensory neurons mediate the thermal acclimatization and that two TRPV channels, OSM-9 and OCR-2, expressed in ADL neurons might contribute to temperature detection relevant for cold tolerance. However, their data suggested that TRPV channels might be accessory and that additional thermosensory molecules were involved. In order to identify these unknown thermo-responsive molecules, the author hypothesized that it could be a GPCR and conducted a RNAi screen covering most GPCR encoded in the worm genome. They identified many GPCRs whose knockdown could either up-regulate or down-regulate cold tolerance. Knocking down *srh-40* with RNAi or with CRISPR-engineered mutations had a strong impact on cold tolerance. They show that SRH-40 is (i) expressed in ADL, (ii) required for ADL to develop normal heat-evoked calcium transients in response to minute-lasting heat stimuli from 13 to 27°C, (iii) able to confer thermal sensitivity to another *C. elegans* thermosensory neuron (ASER) and (iv) able to enhance the pre-existing thermal responsiveness in *Drosophila* S2R+ cells.

The article is well written overall and easy to follow, albeit it only cursorily discusses the results. The RNAi screen is impressive and a great achievement on its own. The data showing the requirement for *srh-40* and the presumable locus of action of SRH-40 in ADL, as well as its role for ADL calcium activity are convincing. In contrast, the dissection of the downstream mechanisms is somewhat confusing. The result of SRH-40 ectopic expression experiments are compatible with the model in which this GPCR is a thermosensory component, but other explanations exist and the data also raise some important questions/doubts. Furthermore, the connection between GPCR and TRPV channel is not addressed appropriately, despite being a very important aspect of the story (and this both regarding the experiments conducted and the discussion). While the presented findings are very interesting and important in the field, some evidence/controls are lacking on some important aspects and I would request that the authors fully address the concerns below, before I can recommend the article for publication.

Response:

We are grateful for the comments from Reviewer #1. We have conducted additional experiments and rewritten the text accordingly.

Major points:

1) Fig. 1f, g,h: It is not clear why the rescue experiments are conducted after 7h at 25°C, while 3h and 5h are used elsewhere. The entire dataset at 3, 5 and 7 h is needed for the reader to evaluate the temporal window for which *srh-40* is relevant in ADL.

Response:

Thank you for pointing this out. We have conducted additional rescue experiments for the abnormal temperature acclimatization of *srh-40* mutants under [15°C→25°C (3 or 5 h)→2°C]. The abnormal increase in temperature acclimatization in *srh-40* mutants under the [15°C→25°C (3 or 5 h)→2°C] condition was rescued by introducing an *srh-40* cDNA driven by an *srh-40* promoter or an ADL-specific promoter. We have added the data in current Fig. 1h and i and moved the graph showing the result of rescue experiments under the [15°C→25°C (7 h)→2°C] condition to Supplementary Fig. 3e because of space limitations.

2) In order to interpret the ASER ectopic expression experiments, we need to have one-to-one comparisons with ADL. We therefore need to see ASER calcium activity in *egl-30* mutant and ADL calcium activity in *goa-1* mutant. The justification brought by the authors to focus on EGL-30 because it could regulate TRPV channel (P3 L35) is hard to follow, because we have no solid reason to believe SRH-40 must work upstream of TRPV channels (neither prior nor after the present study). Of note, *goa-1* and *egl-30* are expressed in both ADL and ASER, based on CENGEN data.

Response:

Thank you for the suggestions and pointing this out. Reviewer #2 has indicated a similar suggestion. We conducted the ASER SRH-40 ectopic expression experiments in the *egl-30* mutant and the ADL calcium imaging in the *goa-1* mutant. We found that the SRH-40-induced temperature responsiveness of ASER was suppressed by the mutation in *egl-30* (Fig. 3d). The ADL of the *goa-1* mutant exhibited normal temperature responses compared with the ADL of wild-type animals (Supplementary Fig. 3g), suggesting that GOA-1 is not involved in ADL temperature sensing. We have discussed the comparison of temperature responses between ADL and ASER in relation to the Gα proteins EGL-30 and GOA-1 and in relation between SRH-40 and TRPV channels. Because Reviewer #2 has raised similar and additional

suggestions, the discussion addresses both concerns. We have rewritten the text and added these results and discussions as follows.

In the Results section ($G_{\alpha q}$ EGL-30 and TRPV act in the ADL temperature signaling.) (P7 L10): “ADL also expresses $G_{\alpha i/o}$ encoded by *goa-1*, but *goa-1* mutant demonstrated normal temperature responses (Supplementary Fig. 3g).”

In the Results section (SRH-40 confers warmth response in taste and culture cells.) (P9 L17): “We observed that the SRH-40-induced temperature responsiveness of ASER was suppressed by the mutation of G_{α} encoded by *goa-1*, *gpa-3* and *egl-30* that are expressed in ASER and ADL thermosensory neurons (Fig. 3c, d).

In the Discussion section (P13 L1): “The ectopic expression of SRH-40 in the ASER neuron conferred warmth sensitivity on the nonwarmth-sensing neuron ASER, and this acquired temperature sensitivity was interfered by the mutation of G_{α} (*goa-1*, *gpa-3*, and *egl-30*), GC (*gcy-5*), and CNGC (*tax-4*) (Fig. 3), suggesting that an SRH-40-dependent thermosensory signal is transduced by these molecules in ASER. Regarding the ADL neuron, the SRH-40-dependent thermosensory signal is transduced by G_{α} EGL-30 and TRPV channels (Fig. 2). These data suggest that CNGCs and TRPV channels are activated through the common G_{α} protein in distinct type of neurons. One such example has been reported in previous study showing olfactory receptor reprogramming⁴¹. An olfactory receptor, ODR-10, expressed in the AWA neuron is required for attraction to diacetyl, and the ectopic expression of ODR-10 in the AWB aversive neuron was found to induce avoidance to diacetyl⁴¹. In both AWA and AWB neurons, the G_{α} protein ODR-3 was commonly required for the transduction of ODR-10-mediated olfactory signaling, and the TRPV channels OSM-9/OCR-2 and CNGCs TAX-2/TAX-4 were required in the respective AWA and AWB neurons⁴¹. Hence, CNGCs and TRP channels are activated by the common G_{α} ODR-3 in the two different types of neurons. Therefore, a set of a receptor and G_{α} could drive multiple downstream signaling pathways depending on components expressed in different cellular contexts. Remarkably, in addition to G_{α} EGL-30, a mutation of another G_{α} *goa-1* interfered the SRH-40-mediated thermal signaling in ASER ectopically expressing SRH-40 (Fig. 3), raising a possibility that multiple types of G_{α} transfer SRH-40-mediated thermal signaling toward CNGCs in ASER.”

3) The connection between TRPV and SRH-40-dependent thermo-responsiveness in ADL is unclear. Data in Fig. 2G, suggest *srh-40* and *ocr-2* might work in the same genetic pathway in regulating ADL warmth-evoked response. But data are not conclusive because of a potential floor effect in the measure, and because *osm-9* was surprisingly left aside of the analysis. Expression in S2R⁺ cells, suggest that SRH-40 has no cumulative effect with OSM-9 OCR-2 in this very artificial system (comparing values in Fig.4 and corresponding expanded data). The following experiments are needed:

- a) ADL calcium imaging in triple *ocr-2 osm-9; srh-40* mutants (ideally the quadruple mutant *ocr-2 osm-9; ocr-1 srh-40*)
- b) If response is still there in the triple, control imaging with a calcium-defective version of cameleon, to demonstrate that residual temperature-evoked response is not an artifact of the sensor.
- c) Survival in *ocr-2 osm-9; srh-40* mutants, compared to *osm-9 ocr-2* double and *srh-40*. In addition, the authors should explicitly what they believe the connection is between TRPV channels and SRH-40 GPCR (notably in the light of the proposed additional experiment results). Early in the manuscript they state that GPCR might modulate TRP channel (notably via EGL-30). Later it looks like it could be two separate pathways. This is very confusing.

Response:

Thank you for pointing this out. As reported previously, *osm-9* and *ocr-2* single mutants and *osm-9 ocr-2* double mutant exhibited abnormal increases in temperature acclimatization under the [15°C→25°C(3 or 5 h)→2°C] condition, and *osm-9* and *ocr-2* single mutants exhibited reduced ADL temperature responses (Ohnishi et al., 2022). However, the *osm-9 ocr-2* double mutant and the *osm-9 ocr-2; ocr-1* triple mutant harboring an additional TRPV mutation exhibited normal ADL temperature responses, possibly because of a compensatory mechanism (Ohnishi et al., 2022). Moreover, *ocr-2* single mutation was more effective than *osm-9* mutation to reduce ADL temperature response (Ohnishi et al., 2022). Therefore, we analyzed an *srh-40; ocr-2* double mutant (Fig. 2i).

According to the suggestion of Reviewer #1, we conducted the following experiments: (1) temperature acclimatization test for double *ocr-2 osm-9* mutant and triple *ocr-2 osm-9; srh-40* mutant.; (2) ADL Ca²⁺ imaging in triple *ocr-2 osm-9; srh-40* mutant and quadruple *ocr-2 osm-9; ocr-1 srh-40* mutant.; (3) Ca²⁺ imaging of ADL using a lower affinity for the calcium version of cameleon YC4.12 instead of YC3.60.

- (1) *srh-40; osm-9 ocr-2* triple mutants demonstrated abnormal increases in temperature acclimatization under the [15°C→25°C→2°C] condition; these phenotypes resemble those of *osm-9 ocr-2* double mutant and greater than that of the *srh-40* mutant (Fig. 2g).
- (2) an *srh-40; osm-9 ocr-2* triple mutant and an *srh-40; osm-9 ocr-2; ocr-1* quadruple mutant exhibited normal ADL temperature responses with Ca²⁺ imaging (Fig. 2h), whose phenotypes were similar to those of *osm-9 ocr-2* double and *osm-9 ocr-2; ocr-1* triple mutants exhibiting a normal phenotype. These phenomena are probably caused by an unidentified compensatory mechanism described in a previous study (Ohnishi et al., 2022).
- (3) We conducted Ca²⁺ imaging of ADL using YC4.12 with a lower Ca²⁺ affinity than YC3.60, and we confirmed that the thermal response of ADL measured by YC4.12 was minimal, suggesting that the temperature-dependent intensity change of YC3.60 in ADL was due to Ca²⁺ increase and not due to an artifact of the cameleon (Fig. 2h).

We conclude that TRPV channels could be downstream of SRH-40 or act in parallel with the SRH-40 pathway. These experimental results and discussion are described in the “Results” and “Discussion” sections of the revised manuscript, as follows.

In the Results section (G_{aq} EGL-30 and TRPV act in the ADL temperature signaling.) (P7 L20): “Previous studies have demonstrated that TRPV mutants exhibit defects in temperature acclimatization and decreased ADL temperature responses, suggesting that TRPV channels are crucial components in ADL temperature sensing^{5,18,21}. Furthermore, *osm-9* and *ocr-2* single mutants and *osm-9 ocr-2* double mutant exhibited abnormal temperature acclimatization under the [15°C→25°C (3 or 5 h)→2°C] condition, and *osm-9* and *ocr-2* single mutants exhibited decreased ADL temperature responses⁵. However, the *osm-9 ocr-2* double mutant and the *osm-9 ocr-2; ocr-1* triple mutant harboring an additional TRPV mutation exhibited normal ADL temperature responses, possibly because of a compensatory mechanism⁵. To explore the functional interaction between SRH-40 and TRPV channels in ADL, we constructed mutants harboring *srh-40* and multiple TRPV channels and evaluated their temperature acclimatization and ADL temperature responses (Fig. 2g–i). We observed that *srh-40; osm-9 ocr-2* triple mutants demonstrated an abnormal increase of survival

rate under the [15°C→25°C→2°C] condition; this phenotype was comparable to that of the *osm-9 ocr-2* double mutant and greater than that of the *srh-40* mutant (Fig. 2g). However, the ADL temperature responses in the *srh-40; osm-9 ocr-2* triple mutant and *srh-40; osm-9 ocr-2; ocr-1* quadruple mutant were indistinguishable from those of wild-type animals (Fig. 2h), which was probably due to an unidentified compensatory mechanism⁵. Because the ADL temperature responses of the TRPV double and triple mutants were not reduced, and the *ocr-2* single mutation was more effective than the *osm-9* mutation in reducing ADL temperature responses⁵, we investigated an *srh-40; ocr-2* double mutant. This mutant exhibited decreased ADL temperature responses (Fig. 2i), a phenotype that was comparable to that of the *ocr-2* and *srh-40* single mutants. We also evaluated cameleon YC4.12 with a lower Ca²⁺ affinity than YC3.60³⁶ and confirmed that the thermal response of ADL was minimal (Fig. 2h), suggesting that the temperature-dependent intensity change of YC3.60 in ADL was due to an increase in Ca²⁺ concentration, but not due to an artifact of the cameleon (Fig. 2h).

Overall, these data suggest that SRH-40, EGL-30, and TRPV channels are crucial components in ADL for temperature acclimatization, and SRH-40 functions upstream of EGL-30. TRPV channels could be downstream of SRH-40 or act in parallel with the SRH-40 pathway.”

4) What is the cause of the temperature-evoked up-steps seen in S2R+ cells in ‘Control’ situation? I note that it is partially decreased by Ca²⁺ washout, which indicates it is not (or at least not entirely) the result of fura-2 being influenced by temperature. Should we conclude that S2R+ cells have already some quite marked intrinsic thermal responsiveness? This must be discussed. In addition, the statement on P5 L4, must be changed from “was evoked by SRH-40” to “was enhanced by SRH-40”.

Response:

Thank you for the suggestion and pointing this out. The heat-induced change observed in the absence of extracellular Ca²⁺ should reflect the temperature-dependent decrease in the K_d value of Fura-2, and the additional Ca²⁺ concentration increase upon heating in control S2R+ cells implies that S2R+ cells are intrinsically temperature-sensitive, and that extracellular Ca²⁺ flows into cells upon heating through an unknown mechanism. Previous deep sequencing analysis showed that few components of thermosensors might be weakly expressed in S2R+ cells. We have added the text in the “Results” and “Discussion” sections of the revised manuscript, as follows.

In the Results section (SRH-40 confers warmth response in taste and culture cells.) (P10 L9): “We observed heat stimulation-induced increases in intracellular Ca^{2+} (Ca^{2+}_i) concentrations in control cells, probably due to the intrinsic temperature sensitivity of a Ca^{2+} indicator Fura-2⁴⁰ and S2R+ cells. In contrast, cells expressing OSM-9/OCR-2 exhibited further increases in Ca^{2+}_i concentration compared with the control during the first and second heat stimulations (Fig. 4, a–e), suggesting that these TRPV channels are responsive to temperature increases as reported previously⁵, and that the Ca^{2+} response of the thermoreceptor is measurable in S2R+ cells.”

In the Discussion section (P13 L22): “We selected *Drosophila* S2R+ cells as the *in vitro* expression system for SRH-40 to evaluate its temperature responsiveness because the culture temperature was equivalent to that for *C. elegans*, and these cells could abundantly express SRH-40. The Ca^{2+}_i concentration in vector-transfected S2R+ cells increased upon heating, whereas the absence of extracellular Ca^{2+} reduced the heat-induced increase in Ca^{2+}_i concentration (Fig. 4). These findings indicate that S2R+ cells are intrinsically temperature-sensitive, and extracellular Ca^{2+} flows into cells upon heating through an unknown mechanism. Deep sequencing analysis has revealed that S2R+ cells scarcely express known temperature sensors such as TRP channels and ionotropic receptors (Irs)⁴². Although Ir25a and Ir93a might be expressed, which are co-receptor and involved in cooling and heating responses⁴³, another component Ir68a was not expressed in S2R+ cells⁴², and there is no evidence indicating that a pair of Ir25a and Ir93a forms a functional heat sensor.”

5) The Control situation in Fig. 4 is not well explained. How could there be only one control dataset, when the control is supposed to be a transfected but un-induced cell preparation? There must be one control per transfected plasmid mix. The total amount of DNA matters a lot for expression in transfection experiments, therefore, proper controls are needed with matching DNA content.

Response:

Thank you for pointing this out. We regret for the insufficient explanation. The total amount of DNA in the transfected plasmid mixture was similar in all experiments in Fig. 4, and the only difference among samples was the presence of indicated components. We have described the details of contained DNA in the “Results”, “legends” and “Methods” of the revised manuscript as follows.

In the Results section (SRH-40 confers warmth response in taste and culture cells.) (P10 L20): “To equalize the transfection condition among samples, we transfected the same amount of a DNA mixture (1.2 μg in total) to cells in all samples.”

In the legend of Fig. 4 (P38 L22): “A DNA mixture of 1.2 µg was transfected to cells under all conditions to equalize the amount of transfection.”

In the Methods section (P23 L11) (Ca²⁺ imaging in *Drosophila* S2R+ cells): “A DNA mixture transfected in the SRH-40 stable cell line shown in Fig. 4 consists of the following compositions: “Control” and “SRH-40” containing 1.0 µg *pAc5.1* vector and 0.2 µg *pAc5.1-mCherry*; “TAX-2, TAX-4” and “SRH-40, TAX-2, TAX-4” containing 0.5 µg *tax-2::T2A::tax-4* in *pAc5.1* vector, 0.5 µg *pAc5.1* vector, and 0.2 µg *pAc5.1-mCherry*; “GOA-1, TAX-2, TAX-4” and “SRH-40, GOA-1, TAX-2, TAX-4” containing 0.5 µg *goa-1* in *pAc5.1* vector, 0.5 µg *tax-2::T2A::tax-4* in *pAc5.1* vector, and 0.2 µg *pAc5.1-mCherry*. “SRH-40, GOA-1” containing 0.5 µg *goa-1* in *pAc5.1* vector, 0.5 µg *pAc5.1* vector and 0.2 µg *pAc5.1-mCherry*; “OSM-9, OCR-2” and “SRH-40, OSM-9, OCR-2” containing 0.5 µg *osm-9::T2A::ocr-2* in *pAc5.1* vector, 0.5 µg *pAc5.1* vector, and 0.2 µg *pAc5.1-mCherry*; “SRH-40, EGL-30, OSM-9, OCR-2” containing 0.5 µg *osm-9::T2A::ocr-2* in *pAc5.1* vector, 0.5 µg *egl-30::T2A::mCherry* in *pMT* vector, and 0.2 µg *pAc5.1* vector.”

6) Generally speaking, it is possible that SRH-40 does not display temperature-dependent G-protein activation activity, but is simply required for the expression, the stabilization or the localization of another thermo-dependent molecule. This must be stated very explicitly and the claims toned down in the entire manuscript.

Response:

We have explicitly added the discussion for SRH-40 functions in temperature acclimatization, and toned down our claims for SRH-40 functions in the entire revised manuscript.

Minor points

1) Fig. 1d brings little in its current form. I guess it is ranked by alphabetical order of the RNAi targets, which is arbitrary and has no helpful meaning. I would suggest to rank the RNAi targets based on the survival rate and highlight *eri-1*; *lin-15B* with an arrow. This will provide a more useful graphic that will not only look nicer but, more importantly, also better present the survival score distribution above and below that of the control.

Response:

Thank you for your suggestion, Reviewer #2 has made a similar suggestion, and we agree with this. We have arranged the RNAi targets in the order of its survival rate, and highlighted *eri-1*; *lin-15B* and *srh-40* with different colors and arrows.

2) P1 L22: “Exhaustive knockdown” should be rephrased (e.g., with “Systematic knockdown”) because not every GPCRs have been targeted.

Response:

Thank you for pointing this out. We have edited the text as suggested.

3) P2 L27: “thermos-receptor” need to be replaced by “thermo-receptor”

Response:

Thank you for indicating this. We have edited the text.

4) Acknowledgements of the CGC should be complemented to follow their guidelines : <https://cgc.umn.edu/acknowledging-the-cgc>

Response:

Thank you for your suggestion. We have described the acknowledgments of the CGC according to the CGC guidelines.

Reviewer #2 (Remarks to the Author):

In this manuscript, the authors (Ohnishi et al.) performed a targeted RNAi screen to find genes that affect cold tolerance in *C. elegans*. From more than one thousand GPCR genes, they found knock-down of *srh-40*, which is highly expressed in the ADL sensory neuron, can increase worms' cold tolerance. Through calcium imaging experiments, they propose *srh-40* as a GPCR-type temperature sensor.

Overall, I think the authors did a lot of nice work, especially on the screen. The *C. elegans* genome contains a very large number of GPCRs. The characterization of the localization of many GPCRs (Extended Data Figure 1) is also very useful to many readers. The gene they found, *srh-40*, is a novel GPCR gene that is only expressed in ADL (CeNGEN). Its knock-down effect on cold tolerance is obvious and quite interesting.

However, I don't think the data supporting *srh-40* as a temperature sensor is strong enough. First of all, the transition from the finding that *srh-40* affects cold tolerance to the hypothesis that SRH-40 is a temperature sensor is abrupt. I understand that SRH-40 is on the membrane and therefore could be suited for the role of a temperature sensor. But GPCRs in neurons can affect cold tolerance in the entire animal in many ways, not necessarily by acting as a thermosensor. Secondly, in all the calcium imaging experiments, the calcium changes due to SRH-40 are quite mild, although significant. This makes it somewhat hard to conclude that SRH-40 is a bona fide thermosensor.

Response:

Thank you for your suggestion, and we agree with this. Reviewer #1 has made a similar suggestion about SRH-40 functions in temperature acclimatization, and we agree with the suggestions of Reviewers #2 and #1. We have toned down our claims for SRH-40 functions in temperature acclimatization in the entire revised manuscript. As the ectopic expression of SRH-40 in the nonwarmth-sensing gustatory neuron ASER and *Drosophila* S2R+ cells resulted in the acquisition of warmth sensitivity, SRH-40 may react directly to temperature changes. However, as Reviewer #2 has indicated, the calcium changes caused by SRH-40 are very mild in the gustatory neuron ASER and *Drosophila* S2R+ cells, so there are other possibilities, such as stabilizing, localizing, and/or facilitating the expression of other thermosensitive molecules associated with temperature acclimation. In that case, the expression of SRH-40 could induce the

expression, stabilization, or localization of other thermosensitive molecules in S2R+ cells, but it occurs only in combination with TAX-2/TAX-4.

We have added the discussion as follows.

In the Discussion (P12 L9): "As the ectopic expression of SRH-40 in the nonwarmth-sensing gustatory neuron ASER and *Drosophila* S2R+ cells resulted in the acquisition of warmth sensitivity, SRH-40 may directly react to temperature changes; however, other possibilities cannot be excluded, such as stabilizing, localizing, and/or facilitating the expression of other thermosensitive molecules that are associated with temperature acclimatization."

In the Discussion (P15 L4): "It is possible that SRH-40 expression induces the expression, stabilization, or localization of other thermosensitive molecules in S2R+ cells; however, but it occurs only in combination with TAX-2/TAX-4."

I also don't understand what the authors try to propose as the signaling pathways downstream from *srh-40*. From the very beginning, the authors mentioned that "OCR-2/OSM-9 may act as accessorial warm receptor downstream of unknown thermos-receptor". They also mentioned that "TRP channels are regulated by Gαq signaling". These statements, fig. 1c, together with the Gαq/*egl-30* found to be involved, all indicate that *srh-40* is the thermosensor upstream from OCR-2/OSM-9 in ADL. However, when authors ectopically express SRH-40 in ASER or *Drosophila* S2 cells, the downstream ion channels are TAX-2/TAX-4 in both cases (ASER only expresses TAX-2/TAX-4, but not the OCR-2/OSM-9 channel). The only explanation I can think of is that *srh-40* can activate both TAX-2/TAX-4 and OCR-2/OSM-9 channels in different contexts. These two channels are very different in nature. I have not known of any upstream player who can activate both. If the authors think *srh-40* or Gαq can activate both channels or know a precedent, it should be discussed.

Response:

Thank you for the suggestions and pointing this out. Reviewer #1 has made a similar suggestion about the analysis of ectopically expressing SRH-40 in ASER, and we agree with the suggestions of Reviewers #2 and #1. We have conducted ASER ectopic expression experiments in the *egl-30* mutant. The SRH-40-induced temperature responsiveness of ASER was suppressed by the mutation in *egl-30* (Fig. 3d). This suggests that *srh-40* can activate both TAX-2/TAX-4 and OCR-2/OSM-9 channels through the same EGL-30 in different contexts, ADL and ASER. We have discussed this result using the report on olfactory receptor reprogramming in *C. elegans* (Troemel et al.,

Cell, 1997). An olfactory receptor, ODR-10, expressed in the AWA neuron is required for attraction to diacetyl, and the ectopic expression of ODR-10 in the AWB aversive neuron induced avoidance to diacetyl. In both AWA and AWB neurons, the G_α protein ODR-3 was commonly required for the transduction of ODR-10-mediated olfactory signaling, and the TRPV channels OSM-9/OCR-2 and CNGCs TAX-2/TAX-4 were required in the respective AWA and AWB neurons (Troemel et al., *Cell*, 1997). Hence, CNGCs and TRP channels are activated by the common G_α ODR-3 in the two different types of neurons. Therefore, a set of a receptor and G_α could drive multiple downstream signaling pathways depending on components expressed in different cellular contexts. In addition to G_α EGL-30, a mutation of another G_α *goa-1* interfered SRH-40-mediated thermal signaling in ASER ectopically expressing SRH-40 (Fig. 3), raising a possibility that multiple types of G_α transfer the SRH-40-mediated thermal signaling toward CNGCs in ASER. We have added the discussion as follows.

In the Discussion section (P13 L1): “The ectopic expression of SRH-40 in the ASER neuron conferred warmth sensitivity on the nonwarmth-sensing neuron ASER, and this acquired temperature sensitivity was interfered by the mutation of G_α (*goa-1*, *gpa-3*, and *egl-30*), GC (*gcy-5*), and CNGC (*tax-4*) (Fig. 3), suggesting that an SRH-40-dependent thermosensory signal is transduced by these molecules in ASER. Regarding the ADL neuron, the SRH-40-dependent thermosensory signal is transduced by G_α EGL-30 and TRPV channels (Fig. 2). These data suggest that CNGCs and TRPV channels are activated through the common G_α protein in distinct type of neurons. One such example has been reported in previous study showing olfactory receptor reprogramming⁴¹. An olfactory receptor, ODR-10, expressed in the AWA neuron is required for attraction to diacetyl, and the ectopic expression of ODR-10 in the AWB aversive neuron was found to induce avoidance to diacetyl⁴¹. In both AWA and AWB neurons, the G_α protein ODR-3 was commonly required for the transduction of ODR-10-mediated olfactory signaling, and the TRPV channels OSM-9/OCR-2 and CNGCs TAX-2/TAX-4 were required in the respective AWA and AWB neurons⁴¹. Hence, CNGCs and TRP channels are activated by the common G_α ODR-3 in the two different types of neurons. Therefore, a set of a receptor and G_α could drive multiple downstream signaling pathways depending on components expressed in different cellular contexts. Remarkably, in addition to G_α EGL-30, a mutation of another G_α *goa-1* interfered the SRH-40-mediated thermal signaling in ASER ectopically expressing

SRH-40 (Fig. 3), raising a possibility that multiple types of G_{α} transfer SRH-40-mediated thermal signaling toward CNGCs in ASER.”

The reason to express “SRH-40, TAX-2 and TAX-4 with or without GOA-1 in S2R+ cells”, but not OCR-2/OSM-9 is that “temperature response of TRPV channels in ADL (OCR-2/OSM-9) were detectable in S2R+”. This reasoning seems problematic to me. The authors propose that SRH-40 and OCR-2/OSM-9 are “dual thermo-sensing machinery”, and the results in ADL indicate that temperature responses from SRH-40 and OCR-2/OSM-9 are additive in ADL. Then, detecting temperature response from OCR-2/OSM-9 in S2R+ cells is not a problem, as long as SRH-40 plus OCR-2/OSM-9 has an increased response than OCR-2/OSM-9 alone, but the results failed to show this increase. The temperature-induced calcium response from OCR-2/OSM-9 is not saturated since adding ionomycin further increased intracellular calcium.

Response:

We are grateful for the comments from Reviewer #2. We have added the discussion and transferred the supplementary text of calcium imaging in S2R+ cells expressing OSM-9/OCR-2 to main text to explain as follows: The heating-induced increases in Ca^{2+}_i concentration in cells expressing OSM-9/OCR-2 with SRH-40 and SRH-40/ $G_{\alpha q}$ EGL-30 were significantly higher than those of the control but comparable to those of cells expressing only OSM-9/OCR-2 (Fig. 4e). This suggested that these components were not sufficient to monitor the temperature response of SRH-40. We then introduced G_{α} GOA-1 and CNGCs TAX-2/TAX-4 into the SRH-40 stable S2R+ cell line, because we could observe a temperature-dependent Ca^{2+} increase with ectopic SRH-40 expression in ASER, and SRH-40 utilized G_{α} GOA-1 and CNGCs TAX-2/TAX-4 to react to warming stimulation in ASER (Fig. 3).

We have added the discussion for the dual thermosensing machinery of SRH-40 and OSM-9/OCR2. We estimated that SRH-40, $G_{\alpha q}$ EGL-30, and OSM-9/OCR-2 cooperatively function to respond to heating in ADL based on our *in vivo* study; however, cells expressing all these components showed no additional increases in Ca^{2+}_i concentration compared with cells expressing only OSM-9/OCR-2. It is probable that unidentified intermediate component(s) between SRH-40 and OSM-9/OCR-2 are missing in S2R+ cells, or that SRH-40 and TRPVs might function independently in temperature sensation in ADL. Irrespective of the mechanism, we propose that both

SRH-40 and OSM-9/OCR-2 are thermosensitive and constitute a dual thermosensing machinery in ADL. We have added the discussion as follows.

In the Discussion section (P14 L10): “We confirmed an increase in Ca^{2+}_i concentration in cells expressing OSM-9/OCR-2 by heat stimulation, which are known to be thermosensitive, suggesting that temperature response is measurable in S2R+ cells. Based on our *in vivo* study, we estimated that SRH-40, $G_{\alpha q}$ EGL-30, and OSM-9/OCR-2 cooperatively function to respond to heating in ADL; however, cells expressing all these components showed no such additional increases in Ca^{2+}_i concentration compared with cells expressing only OSM-9/OCR-2. It is probable that unidentified intermediate component(s) between SRH-40 and OSM-9/OCR-2 are missing in S2R+ cells or that SRH-40 and TRPVs function independently in temperature sensation in ADL. Irrespective of the mechanism, we propose that both SRH-40 and OSM-9/OCR-2 are thermosensitive and constitute a dual thermosensing machinery in ADL.”

The authors propose that SRH-40 and OCR-2/OSM-9 are “dual thermo-sensing machinery”, and the results in ADL indicate that temperature responses from SRH-40 and OCR-2/OSM-9 are additive in ADL

There are other places that, I think, need attention, or that I don't quite understand:

1.1. The cold acclimatization assay that has been performed throughout the manuscript is -- [25°C→15°C→2°C]. During the screen, *srh-40* was found to affect cold tolerance in another protocol -- [20°C→2°C]. This seems to suggest that *srh-40* is playing a role in cold tolerance per se, not necessarily cold acclimatization. This discrepancy has two potential problems. One is whether the mechanism investigated is about intrinsic cold tolerance or cold acclimatization. The other is that the authors probably should do calcium imaging to find out *srh-40*'s role during cooling since the protocol [20°C→2°C] does not involve a warming step. Many genuine thermosensors can only respond to a narrow temperature range. If there is a reason only to look at the response to warming, it should be stated.

Response:

Thank you for pointing this out and the suggestion. We found that two independent *srh-40* knockout mutants, *srh-40(chr41)* and *srh-40(chr42)*, consistently exhibited abnormal temperature acclimatization under the [15°C→25°C (3 or 5 h)→2°C] condition (Fig. 1e-g, Supplementary Fig. 2d-f). However, the *srh-40* knockout

mutants showed no abnormal cold tolerance under the [20°C→2°C] condition (Supplementary Fig. 3d), which was inconsistent with the phenotype of *srh-40* knockdown animals (Fig. 1d). Considering that the expression level of *srh-40* was higher at 25°C than at 15°C (Supplementary Fig. 3c), SRH-40 should be more important for temperature acclimatization to higher temperature but not for cold tolerance. As suggested, we have conducted additional experiments in Ca²⁺ imaging upon cooling. The Ca²⁺ concentration in ADL slightly decreased upon cooling (20°C→10°C) in both wild-type animals and *srh-40* mutants (Fig. 1m). These data suggest that SRH-40 is required for warm rather than cool activation of ADL. We also considered a difference in the phenotype between *srh-40* knockdown animals and *srh-40* knockout mutant. A phenotypic discrepancy between knockout and knockdown could be due to off-target effects of RNAi and/or, e.g., random effectiveness of RNAi among tissues. According to these additional results and discussions, we have rewritten the text accordingly.

In the Results section (Identification of GPCR-based thermoreceptors.) (P4 L2): “Because cold tolerance abnormality was strongly observed in ASJ-defective mutants under the [20°C→2°C] condition (Fig. 1a), we applied the same protocol for this RNAi screening. The survival rate of 20°C-grown *eri-1; lin-15B* animals, the RNAi-sensitive strain, after cold exposure was 28%, which was used as a control for screening (Fig. 1d, Supplementary Data 1). We observed enhanced or reduced cold tolerance in several GPCR genes after the knockdown (Fig. 1d, Supplementary Data 1) and further investigated selected genes based on the order of the strength of the abnormality. We evaluated the expression pattern of 53 of those genes using a GFP reporter and DiI staining and found that 16 genes were expressed in temperature-sensing neurons, including ASJ and ADL, in the head (Supplementary Fig. 1, Supplementary Table 1). We then attempted to introduce mutations in these GPCR genes using CRISPR/Cas9 and evaluated temperature acclimatization of the knockout animals lacking one of 12 GPCR genes under the [15°C→25°C (3 or 5 h)→2°C] condition (Fig. 1b, Supplementary Fig. 2a–f). Among those genes, two independent *srh-40* knockout mutants, *srh-40(chr41)* and *srh-40(chr42)*, consistently exhibited abnormal temperature acclimatization (Fig. 1e–g, Supplementary Fig. 2d–g). *srh-40* expression was detected in the ADL thermosensory neuron (Supplementary Fig. 3a, b), with the expression level being higher at 25°C than at 15°C (Supplementary Fig. 3c). Under the [15°C→2°C] condition, almost all 15°C-grown wild-type and *srh-40* mutant animals survived after cold stimuli

(Fig. 1e). In contrast, when 15°C-grown wild-type animals were maintained at 25°C for 3 or 5 h, ~80% or ~95% of wild-type animals died after cold stimuli, respectively [15°C → 25°C (3 or 5 h) → 2°C] (Fig. 1f, g). However, the survival rates of *srh-40* mutants were significantly higher under the [15°C → 25°C (3 or 5 h) → 2°C] condition (Fig. 1f, g), suggesting that SRH-40 is essential for temperature acclimatization. The *srh-40* knockout mutants did not exhibit abnormal cold tolerance under the [20°C → 2°C] condition (Supplementary Fig. 3d), which was inconsistent with the phenotype of *srh-40* knockdown animals (Fig. 1d). Considering that the expression level of *srh-40* was higher at 25°C than at 15°C (Supplementary Fig. 3c), SRH-40 should be more important for temperature acclimatization but not for cold tolerance. This phenotypic discrepancy between knockout and knockdown could be due to off-target effects of RNAi and/or, e.g., random effectiveness of RNAi among tissues.”

In the Results section (SRH-40 is required for the thermal response of ADL neuron.) (P6 L5): “In contrast, the Ca²⁺ concentration in the ADL slightly decreased upon cooling (20°C → 10°C) in both wild-type and *srh-40* mutants (Fig. 1m). These findings suggest that SRH-40 is essential for warm rather than cool activation of ADL.”

1.2. The authors tested that the *srh-40* null allele did not affect worms chemotaxis to 1-octanol. Given that ADL’s most well-known role is to sense pheromones, I think testing chemotaxis to pheromones, such as C3, could be a more appropriate control.

Response:

We agree with the comments of Reviewer #2. We conducted the avoidance assay using crude aversive pheromone extracted from worms and found that the *srh-40* mutant exhibited normal avoidance behavior against the aversive pheromone. We have rewritten the text as follows:

In the Results section (SRH-40 is required for the thermal response of ADL neuron) (P5 L16): “Studies have reported that ADL detects multiple sensory stimuli, such as aversive odorant 1-octanol, aversive ascaroside pheromones, and temperature^{18,25,33,34}. However, we found that the *srh-40* mutants exhibited normal avoidance against 1-octanol and aversive pheromones (Fig. 1j, k), suggesting that the SRH-40 expressed in ADL is not involved in the avoidance of 1-octanol and aversive pheromones, and that the disabled temperature acclimatization in *srh-40* mutants is not due to a general defect in the neural functions.”

1.3. Whether a decreased calcium response in ADL (as seen in *srh-40* mutants) upon warming in general correlates with a better cold tolerance level. If so, how is it achieved (maybe out of scope)? Why not the opposite (increased calcium response correlates with worse cold tolerance)?

Response:

We agree with the comments of Reviewer #2. The calcium response in ADL upon warming does not necessarily correlate with cold tolerance phenotypes in some mutants. We have added a discussion on the relationship between ADL calcium response and temperature acclimatization by citing a mutant result from a previous study (Okahata et al., 2019).

In the Discussion section (P12 L15): “In the temperature acclimatization assay, the *srh-40* mutant animals exhibited a decreased ADL temperature response and an increased survival rate. A previous study also showed that the *kqt-2* mutant lacking a KCNQ potassium channel exhibited reduced ADL temperature response and abnormal temperature acclimatization²¹, which were equivalent to the phenotypes observed in the *srh-40* mutant. However, the *kqt-3* mutant lacking another KCNQ channel exhibited increases in ADL thermal responsiveness and survival rate in temperature acclimatization²¹. These data, including ours, clearly suggest that ADL activity influences temperature acclimation. Nevertheless, it appears that complex mechanisms that yet remain unidentified exist beyond these controversies, which requires further research.”

Some minor issues:

2.1. In fig. 1d, the X axis is better organized according to the order of its survival rate. The control is somewhere in the middle. Then, use different colors to highlight the control and *srh-40*. This gives the readers a sense of where *srh-40* is placed on the entire screen.

Response:

Thank you for your suggestion, Reviewer #1 has raised a similar concern, and we agree with this it. We have arranged the RNAi targets in the order of its survival rate and highlighted *eri-1*; *lin-15B* and *srh-40* with different colors and arrows.

2.2. On page 5, line 2, the statement “the Ca²⁺ increase disappeared” is not accurate. I think the calcium increase was dampened, but it did not disappear.

Response:

Thank you for indicating this. We have edited the text as “the increase in Ca²⁺_i concentration diminished in cells”.

2.3. There is no discussion. The last paragraph merely summarized their findings. I think the manuscript has a very good start, and finding that SRH-40 affecting cold tolerance is very cool. I also think the mechanism by which SRH-40 affects cold tolerance should be looked into more carefully.

Response:

Thank you for your suggestion. We have carefully constructed the discussion for SRH-40 functions in temperature acclimatization and temperature signaling in ADL. We have added the discussion in the revised manuscript as follows.

In the Discussion section (P12 L6): “Our study describes the function of GPCR SRH-40 as a novel thermosensor in the temperature acclimatization of *C. elegans*. Based on genetic and Ca²⁺ imaging analyses, our results elucidated that SRH-40 and downstream G_α signaling are required for ADL thermosensation and subsequent temperature acclimatization. As the ectopic expression of SRH-40 in the nonwarmth-sensing gustatory neuron ASER and *Drosophila* S2R+ cells resulted in the acquisition of warmth sensitivity, SRH-40 may directly react to temperature changes; however, other possibilities cannot be excluded, such as stabilizing, localizing, and/or facilitating the expression of other thermosensitive molecules that are associated with temperature acclimatization.

In the temperature acclimatization assay, the *srh-40* mutant animals exhibited a decreased ADL temperature response and an increased survival rate. A previous study also showed that the *kqt-2* mutant lacking a KCNQ potassium channel exhibited reduced ADL temperature response and abnormal temperature acclimatization²¹, which were equivalent to the phenotypes observed in the *srh-40* mutants. However, the *kqt-3* mutant lacking another KCNQ channel exhibited increases in ADL thermal responsiveness and survival rate in temperature acclimatization²¹. These data, including ours, clearly suggest that ADL activity influences temperature acclimatization. Nevertheless, it appears that complex mechanisms that yet remain unidentified exist beyond these controversies, which requires further research.

The ectopic expression of SRH-40 in the ASER neuron conferred warmth sensitivity on the nonwarmth-sensing neuron ASER, and this acquired temperature sensitivity was interfered by the mutation of G_{α} (*goa-1*, *gpa-3*, and *egl-30*), GC (*gcy-5*), and CNGC (*tax-4*) (Fig. 3), suggesting that an SRH-40-dependent thermosensory signal is transduced by these molecules in ASER. Regarding the ADL neuron, the SRH-40-dependent thermosensory signal is transduced by G_{α} EGL-30 and TRPV channels (Fig. 2). These data suggest that CNGCs and TRPV channels are activated through the common G_{α} protein in distinct type of neurons. One such example has been reported in previous study showing olfactory receptor reprogramming⁴¹. An olfactory receptor, ODR-10, expressed in the AWA neuron is required for attraction to diacetyl, and the ectopic expression of ODR-10 in the AWB aversive neuron was found to induce avoidance to diacetyl⁴¹. In both AWA and AWB neurons, the G_{α} protein ODR-3 was commonly required for the transduction of ODR-10-mediated olfactory signaling, and the TRPV channels OSM-9/OCR-2 and CNGCs TAX-2/TAX-4 were required in the respective AWA and AWB neurons⁴¹. Hence, CNGCs and TRP channels are activated by the common G_{α} ODR-3 in the two different types of neurons. Therefore, a set of a receptor and G_{α} could drive multiple downstream signaling pathways depending on components expressed in different cellular contexts. Remarkably, in addition to G_{α} EGL-30, a mutation of another G_{α} *goa-1* interfered the SRH-40-mediated thermal signaling in ASER ectopically expressing SRH-40 (Fig. 3), raising a possibility that multiple types of G_{α} transfer SRH-40-mediated thermal signaling toward CNGCs in ASER.

We selected *Drosophila* S2R+ cells as the *in vitro* expression system for SRH-40 to evaluate its temperature responsiveness because the culture temperature was equivalent to that for *C. elegans*, and these cells could abundantly express SRH-40. The Ca^{2+}_i concentration in vector-transfected S2R+ cells increased upon heating, whereas the absence of extracellular Ca^{2+} reduced the heat-induced increase in Ca^{2+}_i concentration (Fig. 4). These findings indicate that S2R+ cells are intrinsically temperature-sensitive, and extracellular Ca^{2+} flows into cells upon heating through an unknown mechanism. Deep sequencing analysis has revealed that S2R+ cells scarcely express known temperature sensors such as TRP channels and ionotropic receptors (Irs)⁴². Although Ir25a and Ir93a might be expressed, which are co-receptor and involved in cooling and heating responses⁴³, another component Ir68a was not expressed in S2R+ cells⁴², and there is no evidence indicating that a pair of Ir25a and Ir93a forms a functional heat sensor.

We confirmed an increase in Ca^{2+}_i concentration in cells expressing OSM-9/OCR-2 by heat stimulation, which are known to be thermosensitive, suggesting that temperature response is measurable in S2R+ cells. Based on our *in vivo* study, we determined that SRH-40, $G_{\alpha q}$ EGL-30, and OSM-9/OCR-2 cooperatively function to respond to heating in ADL; however, cells expressing all these components did not show the additional increases in Ca^{2+}_i concentration compared with cells expressing only OSM-9/OCR-2. It is probable that unidentified intermediate component(s) between SRH-40 and OSM-9/OCR-2 are missing in S2R+ cells or that SRH-40 and TRPVs function independently in temperature sensation in ADL. Irrespective of the mechanism, we propose that both SRH-40 and OSM-9/OCR-2 are thermosensitive and constitute a dual thermosensing machinery in ADL.

The concentration of Ca^{2+}_i in cells expressing SRH-40/TAX-2/TAX-4 significantly increased upon heating (Fig. 4). Importantly, there was no such increase in Ca^{2+}_i concentration when cells expressed any other combinations, including SRH-40 alone, TAX-2/TAX-4, SRH-40/GOA-1, or GOA-1/TAX-2/TAX-4. These data strongly suggest that SRH-40 and CNGCs are vital components for temperature response in S2R+ cells and are consistent with the evidence that ectopically expressing SRH-40 in warmth-insensitive ASER acquired thermal responsiveness through G_{α} and CNGCs, and that the ADL temperature response was driven by SRH-40. It is possible that SRH-40 expression induces the expression, stabilization, or localization of other thermosensitive molecules in S2R+ cells; however, it occurs only in combination with TAX-2/TAX-4.

The findings of this study provide a series of evidence that GPCRs act as thermosensors in temperature signaling in the thermosensory neuron, regulating temperature acclimatization in animals. Together with our previous data indicating that the TRPV channels OSM-9/OCR-2 are temperature-sensitive in the ADL neuron^{5,18,21}, we propose that SRH-40 and TRPV channels constitute a dual thermosensing machinery in a single sensory neuron, functioning either in parallel or independently. Another GPCR, rhodopsin, has been reported to contribute to thermotaxis in *Drosophila*^{6,8,44} and mammalian sperm⁴⁵, suggesting that the GPCR-mediated temperature sensation could be evolutionarily conserved. Our study may provide insights into clarifying the mechanisms underlying the temperature sensitivity of GPCRs across species."

REVIEWER COMMENTS

Reviewer #1 (Remarks to the Author):

The manuscript has been significantly modified to include new results and clarifications, which have addressed my concerns. IMHO, the main claims of the article are well supported now. The work will be very important in the field and more broadly, improving our understanding of thermosensation and contributing to go beyond the 'classical TRP-centric' view.

Congratulations to the authors for this terrific job.

Reviewer #2 (Remarks to the Author):

I appreciate the efforts made in this revision, and it has addressed most of my concerns. However, by looking carefully into the data, I still have the following questions that need to be addressed:

1. Please revise the subtitle "Identification of GPCR-based thermoreceptors." I think we are clear that SRH-40 may not be a bona fide thermoreceptor. Please also change any statements throughout the text that claim SRH-40 as a thermoreceptor.

2. I find the conclusion that "The *srh-40* knockout mutants did not exhibit abnormal cold tolerance under the [20°C→2°C] condition" to be problematic. Given that all three genotypes pretty much all died after treatment (Fig. S3c), it is impossible to make meaningful comparisons of cold tolerance level among them. Similar problem with Fig. 1e. In light of this, the explanation "This phenotypic discrepancy between knockout and knockdown could be due to off-target effects of RNAi" appears somewhat too convenient. If that is true, then people can argue the entire RNAi screen is not valid. I propose the following experiments to address these concerns:

a. Treatment of [20°C→2°C] and [15°C→2°C] for a proper duration, aiming to achieve a survival rate close to 50% in the WT. Then compare the survival rate of the mutants. If similar to WT, then the authors can conclude that *srh-40* knockouts have similar cold tolerance level as WT.

b. Perform QPCR experiment to measure the extent of *srh-40* knocked-down in RNAi treatment at 20°C.

3. I've noticed that the quality of the pictures in the PDF file seems to have a very bad quality. When zoomed to 200%, all the figures appear quite blurry. This issue does not seem to be present from the initial submission. I am not sure whether it is just display on my end or an issue with the file itself.

Some minor issues:

1. Change "nonwarmth-sensing" to "non-warmth-sensing" or something better.
2. Include PMID: 18667708 in the citation for the statement "AWC; however, the thermoreceptor has not been identified till date".
3. In the Methods, specify whether the data points in the scatter plot for YFP/CFP % intensity represent the highest point in the calcium trace or a data point at a specific time.
4. Consider merging panels a and c, as well as b and d, in Fig. 4a-d, with lines indicating the enlargement from a to c and b to d.
5. Modify the sentence "SRH-40 confers warmth response in taste and culture cells". At least change it to "cultured cells".
6. Explain why "TAX-4 may be an effector responsible for the increase in Ca²⁺ concentration". On the contrary, CMK-1 may be an effector, but not TAX-4.
7. Address the unusual decrease in calcium traces in Fig. 4d upon the initial temperature decrease from 25°C to 15°C, which is not seen in other panels in Fig. 4. Since neuron inhibition is equally important as activation, this needs to be explained.
8. Explain Fig. 1d where most GPCR gene knock-down increased cold tolerance during the RNAi screen, causing the control to be in the far right. Offer speculations or potential reasons for this phenomenon.

RESPONSE TO REVIEWERS

Reviewer #1 (Remarks to the Author):

The manuscript has been significantly modified to include new results and clarifications, which have addressed my concerns. IMHO, the main claims of the article are well supported now. The work will be very important in the field and more broadly, improving our understanding of thermosensation and contributing to go beyond the 'classical TRP-centric' view.

Congratulations to the authors for this terrific job.

Response:

Thank you for your positive feedback. We are grateful that you believe in the significance of our work.

Reviewer #2 (Remarks to the Author):

I appreciate the efforts made in this revision, and it has addressed most of my concerns. However, by looking carefully into the data, I still have the following questions that need to be addressed:

1. Please revise the subtitle "Identification of GPCR-based thermoreceptors." I think we are clear that SRH-40 may not be a bona fide thermoreceptor. Please also change any statements throughout the text that claim SRH-40 as a thermoreceptor.

Response:

Thank you for the suggestions, we have edited the text including main title and subtitle.

2. I find the conclusion that "The srh-40 knockout mutants did not exhibit abnormal cold tolerance under the [20°C→2°C] condition" to be problematic. Given that all three genotypes pretty much all died after treatment (Fig. S3c), it is impossible to make meaningful comparisons of cold tolerance level among them. Similar problem with Fig. 1e. In light of this, the explanation "This phenotypic discrepancy between knockout and knockdown could be due to off-target effects of RNAi" appears somewhat too convenient. If that is true, then people can argue the entire RNAi screen is not valid. I propose the following experiments to address these concerns:

a. Treatment of [20°C→2°C] and [15°C→2°C] for a proper duration, aiming to achieve a survival rate close to 50% in the WT. Then compare the survival rate of the mutants. If similar to WT, then the authors can conclude that *srh-40* knockouts have similar cold tolerance level as WT.

b. Perform QPCR experiment to measure the extent of *srh-40* knocked-down in RNAi treatment at 20°C.

Response:

Thank you for pointing this out. Following the recommendation of Reviewer #2, we conducted the cold tolerance test and qPCR experiment. *srh-40* mutants exhibited normal cold tolerance under [20°C→2°C (5–7 h)] or [15°C→2°C (72 h)] conditions, where the wild-type exhibited 50% survival rate (Supplementary Fig. 3d, f). We confirmed that WT and *srh-40* knockouts exhibited similar cold tolerance. Our qPCR analysis indicated a significant reduction in *srh-40* gene expression levels in the *srh-40* knockdown animal (*eri-1; lin-15B* background) (Supplementary Fig. 3g). We incorporated these results into the revised manuscript.

3. I've noticed that the quality of the pictures in the PDF file seems to have a very bad quality. When zoomed to 200%, all the figures appear quite blurry. This issue does not seem to be present from the initial submission. I am not sure whether it is just display on my end or an issue with the file itself.

Response:

In this revised manuscript, we embedded high quality images of figures in the manuscript and provided high resolution figures in as a separated PDF to solve the issue.

Some minor issues:

1. Change "nonwarmth-sensing" to "non-warmth-sensing" or something better.

Response:

Thank you for indicating this, we have changed the text.

2. Include PMID: 18667708 in the citation for the statement "AWC; however, the thermoreceptor has not been identified till date".

Response:

Thank you for indicating this, we have added this reference.

3. In the Methods, specify whether the data points in the scatter plot for YFP/CFP % intensity

represent the highest point in the calcium trace or a data point at a specific time.

Response:

Thank you for indicating this. This information has been provided in the figure legend. According to the suggestion of Reviewer #2, we added a similar explanation in the method section.

4. Consider merging panels a and c, as well as b and d, in Fig. 4a-d, with lines indicating the enlargement from a to c and b to d.

Response:

Thank you for indicating this, we have rearranged the figure 4.

5. Modify the sentence "SRH-40 confers warmth response in taste and culture cells". At least change it to "cultured cells".

Response:

Thank you for indicating this, we have changed the text.

6. Explain why "TAX-4 may be an effector responsible for the increase in Ca^{2+} concentration". On the contrary, CMK-1 may be an effector, but not TAX-4.

Response:

Thank you for indicating this. We changed the terms "an effector" to "a primary channel".

7. Address the unusual decrease in calcium traces in Fig. 4d upon the initial temperature decrease from 25°C to 15°C, which is not seen in other panels in Fig. 4. Since neuron inhibition is equally important as activation, this needs to be explained.

Response:

Thank you for pointing this out. We analyzed the Ca^{2+} imaging data sets and confirmed that the extent of decrease in Ca^{2+}_i concentration in response to the cooling stimulation was not different between control and the cells expressing OSM-9/OCR-2 (see figures below). Based on this information, we substituted the representative traces in the revised Figure 4 with those showing typical fluctuations in Ca^{2+}_i concentration.

Comparison of the minimum decrease from baseline (25°C) in Ca^{2+}_i concentration in response to cooling stimulation ($\sim 15^\circ\text{C}$) in the control (vector transfection, no CuSO_4 induction) and cells expressing OSM-9/OCR-2 (highlighted in green in bottom of Fig. 4a, b).

8. Explain Fig. 1d where most GPCR gene knock-down increased cold tolerance during the RNAi screen, causing the control to be in the far right. Offer speculations or potential reasons for this phenomenon.

Response:

Thank you for the suggestions. There is evidence that the majority of cold tolerance mutants isolated previously exhibit abnormal increases, rather than decreases, in cold tolerance (Ohta et al., 2014; Ujisawa et al., 2018; Motomura et al., 2022). For example, *unc-104* mutant, impairing a kinesin in most neurons, shows an abnormal increase in cold tolerance (Ohta et al., 2014), suggesting that a defect in the entire nervous system results in an increase in cold tolerance. Additionally, abnormalities in other tissues such as muscle, intestine, and sperm lead to an abnormal increase in cold tolerance (Ohta et al., 2014 ; Ujisawa et al., 2018; Sonoda et al., 2016; Motomura et al., 2022). Since GPCRs are expressed in various tissues, it is speculated that knocking down most GPCR genes induces abnormal increases in cold tolerance. We added the following sentences in the manuscript:

In the Discussion section (P12 L10): “In RNAi screening, the knockdown of numerous GPCR genes in animals were consistently associated with an elevation in cold tolerance. This observation aligns with the previous studies, where the majority of mutants isolated previously exhibited abnormal increases, rather than decrease, in cold tolerance^{15,19-21,42}. For example, the *unc-104* mutant, impairing a kinesin in most neurons, displayed an abnormal increase in cold tolerance¹⁵. This implies that a defect in the entire

nervous system leads to an enhanced cold tolerance. Moreover, abnormalities in various tissues such as muscle, intestine, and sperm were associated with abnormal increases in cold tolerance^{15,19-21,42}. Given that GPCRs are expressed in diverse tissues, it is hypothesized that knockdown of GPCR genes may exhibit a tendency toward abnormal increases in cold tolerance.”

REVIEWERS' COMMENTS

Reviewer #2 (Remarks to the Author):

I have thoroughly reviewed the most recently updated manuscript and am pleased to note that the authors have demonstrated a commendable understanding of my concerns, addressing each of them successfully. The conclusions drawn are well-supported, and I am in favor of its publication.

Reviewer #2 (Remarks to the Author):

I have thoroughly reviewed the most recently updated manuscript and am pleased to note that the authors have demonstrated a commendable understanding of my concerns, addressing each of them successfully. The conclusions drawn are well-supported, and I am in favor of its publication.

Response:

Thank you for your positive feedback. We are grateful that you believe in the significance of our work.